# Novel risk loci for COVID-19 hospitalization among admixed American populations

Silvia Diz-de Almeida[1,2,3,4†], Raquel Cruz[1,2,3,4†], Andre D Luchessi[5],
José M Lorenzo-Salazar[6], Miguel López de Heredia[3], Inés Quintela[7],
Rafaela González-Montelongo[6], Vivian Nogueira Silbiger[5], Marta Sevilla Porras[3,8],
Jair Antonio Tenorio Castaño[1,3,8], Julian Nevado[1,3,8], Jose María Aguado[9,10,11],
Carlos Aguilar[12], Sergio Aguilera-Albesa[2,13], Virginia Almadana[14],
Berta Almoguera[3,15], Nuria Alvarez[16], Álvaro Andreu-Bernabeu[17,18],
Eunate Arana-Arri[19,20], Celso Arango[17,18,21], María J Arranz[22], Maria-Jesus Artiga[23],
Raúl C Baptista-Rosas[24,25,26], María Barreda- Sánchez[27,28],
Moncef Belhassen-Garcia[29], Joao F Bezerra[30], Marcos AC Bezerra[31],
Lucía Boix-Palop[32], María Brion[33,34], Ramón Brugada[34,35,36,37], Matilde Bustos[38],
Enrique J Calderón[38,39,40], Cristina Carbonell[41,42], Luis Castano[3,19,43,44,45],
Jose E Castelao[46], Rosa Conde-Vicente[47], M Lourdes Cordero-Lorenzana[48],
Jose L Cortes-Sanchez[49,50], Marta Corton[3,15], M Teresa Darnaude[51],
Alba De Martino-Rodríguez[52,53], Victor del Campo-Pérez[54],
Aranzazu Diaz de Bustamante[51], Elena Domínguez-Garrido[55], Rocío Eirós[56],
María Carmen Fariñas[57,58,59], María J Fernandez-Nestosa[60],
Uxía Fernández-Robelo[61], Amanda Fernández-Rodríguez[11,62],
Tania Fernández-Villa[40,63], Manuela Gago-Dominguez[7,64], Belén Gil-Fournier[65],
Javier Gómez-Arrue[52,53], Beatriz González Álvarez[52,53],
Fernan Gonzalez Bernaldo de Quirós[66], Anna González-Neira[16],
Javier González-Peñas[17,18,21], Juan F Gutiérrez-Bautista[67], María José Herrero[68,69],
Antonio Herrero-Gonzalez[70], María A Jimenez-Sousa[11,62], María Claudia Lattig[71,72],
Anabel Liger Borja[73], Rosario Lopez-Rodriguez[3,15,74], Esther Mancebo[75,76],
Caridad Martín-López[73], Vicente Martín[40,63], Oscar Martinez-Nieto[72,77],
Iciar Martinez-Lopez[78,79], Michel F Martinez-Resendez[49], Angel Martinez-Perez[80],
Juliana F Mazzeu[81,82,83], Eleuterio Merayo Macías[84], Pablo Minguez[3,15],
Victor Moreno Cuerda[85,86], Silviene F Oliveira[83,87,88,89], Eva Ortega-Paino[23],
Mara Parellada[17,18,21], Estela Paz-Artal[75,76,90], Ney PC Santos[91],
Patricia Pérez-Matute[92], Patricia Perez[93], M Elena Pérez-Tomás[28], Teresa Perucho[94],
Mellina Pinsach-Abuin[34,35], Guillermo Pita[16], Ericka N Pompa-Mera[95,96],
Gloria L Porras-Hurtado[97], Aurora Pujol[3,98,99], Soraya Ramiro León[65],
Salvador Resino[11,62], Marianne R Fernandes[91,100], Emilio Rodríguez-Ruiz[64,101],
Fernando Rodriguez-Artalejo[40,102,103,104], José A Rodriguez-Garcia[105],
Francisco Ruiz-Cabello[64,106,107], Javier Ruiz-Hornillos[108,109,110], Pablo Ryan[11,111,112,113],
José Manuel Soria[80], Juan Carlos Souto[114], Eduardo Tamayo[115,116],
Alvaro Tamayo-Velasco[117], Juan Carlos Taracido-Fernandez[70], Alejandro Teper[118],
Lilian Torres-Tobar[119], Miguel Urioste[120], Juan Valencia-Ramos[121], Zuleima Yáñez[122],
Ruth Zarate[123], Itziar de Rojas[124,125], Agustín Ruiz[124,125], Pascual Sánchez[126],
Luis Miguel Real[127], SCOURGE Cohort Group, Encarna Guillen-Navarro[28,128,129,130],
Carmen Ayuso[3,15], Esteban Parra[131], José A Riancho[3,57,58,59],
Augusto Rojas-Martinez[132], Carlos Flores[6,133,134,135]*, Pablo Lapunzina[1,3,8]*,
Ángel Carracedo[3,4,7,64]*

*For correspondence:
cflores@ull.edu.es (CF);
angel.carracedo@usc.es (ÁC)

†These authors contributed
equally to this work

Competing interest: The authors
declare that no competing
interests exist.

Reviewing Editor: Siming Zhao,
Dartmouth College, United
States

[1]ERN-ITHACA-European Reference Network, Soria, Spain; [2]Pediatric Neurology Unit, Department of Pediatrics, Navarra Health Service Hospital, Pamplona, Spain; [3]CIBERER, ISCIII, Madrid, Spain; [4]Centro Singular de Investigación en Medicina Molecular y Enfermedades Crónicas (CIMUS), Universidade de Santiago de Compostela, Santiago de Compostela, Spain; [5]Universidade Federal do Rio Grande do Norte, Departamento de Analises Clinicas e Toxicologicas, Natal, Brazil; [6]Genomics Division, Instituto Tecnológico y de Energías Renovables, Santa Cruz de Tenerife, Spain; [7]Fundación Pública Galega de Medicina Xenómica, Sistema Galego de Saúde (SERGAS), Santiago de Compostela, Spain; [8]Instituto de Genética Médica y Molecular (INGEMM), Hospital Universitario La Paz IDIPAZ, Madrid, Spain; [9]Unit of Infectious Diseases, Hospital Universitario 12 de Octubre, Instituto de Investigación Sanitaria Hospital 12 de Octubre (imas12), Madrid, Spain; [10]Spanish Network for Research in Infectious Diseases (REIPI RD16/0016/0002), Instituto de Salud Carlos III, Madrid, Spain; [11]CIBERINFEC, ISCIII, Madrid, Spain; [12]Hospital General Santa Bárbara de Soria, Soria, Spain; [13]Navarra Health Service, NavarraBioMed Research Group, Pamplona, Spain; [14]Hospital Universitario Virgen Macarena, Neumología, Sevilla, Spain; [15]Department of Genetics & Genomics, Instituto de Investigación Sanitaria-Fundación Jiménez Díaz University Hospital - Universidad Autónoma de Madrid (IIS-FJD, UAM), Madrid, Spain; [16]Spanish National Cancer Research Centre, Human Genotyping-CEGEN Unit, Madrid, Spain; [17]Department of Child and Adolescent Psychiatry, Institute of Psychiatry and Mental Health, Hospital General Universitario Gregorio Marañón (IiSGM), Madrid, Spain; [18]School of Medicine, Universidad Complutense, Madrid, Spain; [19]Biocruces Bizkai HRI, Bizkaia, Spain; [20]Cruces University Hospital, Osakidetza, Bizkaia, Spain; [21]Centre for Biomedical Network Research on Mental Health (CIBERSAM), Instituto de Salud Carlos III, Madrid, Spain; [22]Fundació Docència I Recerca Mutua Terrassa, Barcelona, Spain; [23]Spanish National Cancer Research Centre, CNIO Biobank, Madrid, Spain; [24]Hospital General de Occidente, Zapopan Jalisco, Mexico; [25]Centro Universitario de Tonalá, Universidad de Guadalajara, Tonalá Jalisco, Mexico; [26]Centro de Investigación Multidisciplinario en Salud, Universidad de Guadalajara, Tonalá Jalisco, Mexico; [27]Universidad Católica San Antonio de Murcia (UCAM), Murcia, Spain; [28]Instituto Murciano de Investigación Biosanitaria (IMIB-Arrixaca), Murcia, Spain; [29]Hospital Universitario de Salamanca-IBSAL, Servicio de Medicina Interna-Unidad de Enfermedades Infecciosas, Salamanca, Spain; [30]Escola Tecnica de Saúde, Laboratorio de Vigilancia Molecular Aplicada, Brasilia, Brazil; [31]Federal University of Pernambuco, Genetics Postgraduate Program, Recife, Brazil; [32]Hospital Universitario Mutua Terrassa, Barcelona, Spain; [33]Instituto de Investigación Sanitaria de Santiago (IDIS), Xenética Cardiovascular, Santiago de Compostela, Spain; [34]CIBERCV, ISCIII, Madrid, Spain; [35]Cardiovascular Genetics Center, Institut d'Investigació Biomèdica Girona (IDIBGI), Girona, Spain; [36]Medical Science Department, School of Medicine, University of Girona, Girona, Spain; [37]Hospital Josep Trueta, Cardiology Service, Girona, Spain; [38]Institute of Biomedicine of Seville (IBiS), Consejo Superior de Investigaciones Científicas (CSIC)- University of Seville- Virgen del Rocio University Hospital, Seville, Spain; [39]Departamento de Medicina, Hospital Universitario Virgen del Rocío, Universidad de Sevilla, Seville, Spain; [40]CIBERESP, ISCIII, Madrid, Spain; [41]Hospital Universitario de Salamanca-IBSAL, Servicio de Medicina Interna, Salamanca, Spain; [42]Universidad de Salamanca, Salamanca, Spain; [43]Osakidetza, Cruces University Hospital, Bizkaia, Spain; [44]Centre for Biomedical Network Research on Diabetes and Metabolic Associated Diseases (CIBERDEM), Instituto de Salud Carlos III, Madrid, Spain; [45]University of Pais Vasco, UPV/EHU, Bizkaia, Spain; [46]Oncology and Genetics Unit, Instituto de Investigacion Sanitaria Galicia Sur, Xerencia de Xestion Integrada de Vigo-Servizo Galego de

Saúde, Vigo, Spain; [47]Hospital Universitario Río Hortega, Valladolid, Spain; [48]Servicio de Medicina intensiva, Complejo Hospitalario Universitario de A Coruña (CHUAC), Sistema Galego de Saúde (SERGAS), A Coruña, Spain; [49]Tecnológico de Monterrey, Monterrey, Mexico; [50]Department of Microgravity and Translational Regenerative Medicine, Otto von Guericke University, Magdeburg, Germany; [51]Hospital Universitario Mostoles, Unidad de Genética, Madrid, Spain; [52]Instituto Aragonés de Ciencias de la Salud (IACS), Zaragoza, Spain; [53]Instituto Investigación Sanitaria Aragón (IIS-Aragon), Zaragoza, Spain; [54]Preventive Medicine Department, Instituto de Investigacion Sanitaria Galicia Sur, Xerencia de Xestion Integrada de Vigo-Servizo Galego de Saúde, Vigo, Spain; [55]Unidad Diagnóstico Molecular, Fundación Rioja Salud, La Rioja, Spain; [56]Hospital Universitario de Salamanca-IBSAL, Servicio de Cardiología, Salamanca, Spain; [57]IDIVAL, Cantabria, Spain; [58]Hospital U M Valdecilla, Cantabria, Spain; [59]Universidad de Cantabria, Cantabria, Spain; [60]Universidad Nacional de Asunción, Facultad de Politécnica, Paraguay, United States; [61]Urgencias Hospitalarias, Complejo Hospitalario Universitario de A Coruña (CHUAC), Sistema Galego de Saúde (SERGAS), A Coruña, Spain; [62]Unidad de Infección Viral e Inmunidad, Centro Nacional de Microbiología (CNM), Instituto de Salud Carlos III (ISCIII), Madrid, Spain; [63]Grupo de Investigación en Interacciones Gen-Ambiente y Salud (GIIGAS) - Instituto de Biomedicina (IBIOMED), Universidad de León, León, Spain; [64]IDIS, Seongnam, Republic of Korea; [65]Hospital Universitario de Getafe, Servicio de Genética, Madrid, Spain; [66]Ministerio de Salud Ciudad de Buenos Aires, Buenos Aires, Argentina; [67]Hospital Universitario Virgen de las Nieves, Servicio de Análisis Clínicos e Inmunología, Granada, Spain; [68]IIS La Fe, Plataforma de Farmacogenética, Valencia, Spain; [69]Universidad de Valencia, Departamento de Farmacología, Valencia, Spain; [70]Data Analysis Department, Instituto de Investigación Sanitaria-Fundación Jiménez Díaz University Hospital - Universidad Autónoma de Madrid (IIS-FJD, UAM), Madrid, Spain; [71]Universidad de los Andes, Facultad de Ciencias, Bogotá, Colombia; [72]SIGEN Alianza Universidad de los Andes - Fundación Santa Fe de Bogotá, Bogotá, Colombia; [73]Hospital General de Segovia, Medicina Intensiva, Segovia, Spain; [74]Facultad de Farmacia, Universidad San Pablo-CEU, CEU Universities, Urbanización Montepríncipe, Boadilla del Monte, Spain; [75]Hospital Universitario 12 de Octubre, Department of Immunology, Madrid, Spain; [76]Instituto de Investigación Sanitaria Hospital 12 de Octubre (imas12), Transplant Immunology and Immunodeficiencies Group, Madrid, Spain; [77]Fundación Santa Fe de Bogota, Departamento Patologia y Laboratorios, Bogotá, Colombia; [78]Unidad de Genética y Genómica Islas Baleares, Islas Baleares, Spain; [79]Hospital Universitario Son Espases, Unidad de Diagnóstico Molecular y Genética Clínica, Islas Baleares, Spain; [80]Genomics of Complex Diseases Unit, Research Institute of Hospital de la Santa Creu i Sant Pau, IIB Sant Pau, Barcelona, Spain; [81]Universidade de Brasília, Faculdade de Medicina, Brasília, Brazil; [82]Programa de Pós-Graduação em Ciências Médicas (UnB), Brasília, Brazil; [83]Programa de Pós-Graduação em Ciencias da Saude (UnB), Brazila, Brazil; [84]Hospital El Bierzo, Unidad Cuidados Intensivos, León, Spain; [85]Hospital Universitario Mostoles, Medicina Interna, Madrid, Spai, Spain; [86]Universidad Francisco de Vitoria, Madrid, Spain; [87]Departamento de Genética e Morfologia, Instituto de Ciências Biológicas, Universidade de Brasília, Brasília, Brazil; [88]Programa de Pós-Graduação em Biologia Animal (UnB), Brasília, Brazil; [89]Programa de Pós-Graduação Profissional em Ensino de Biologia (UnB), Brasília, Brazil; [90]Universidad Complutense de Madrid, Department of Immunology, Ophthalmology and ENT, Madrid, Spain; [91]Universidade Federal do Pará, Núcleo de Pesquisas em Oncologia, Belém, Brazil; [92]Infectious Diseases, Microbiota and Metabolism Unit, CSIC Associated Unit, Center for Biomedical Research of La Rioja (CIBIR), Logroño, Spain; [93]Inditex, A Coruña,

Corunna, Spain; [94]GENYCA, Madrid, Spain; [95]Instituto Mexicano del Seguro Social (IMSS), Centro Médico Nacional Siglo XXI, Unidad de Investigación Médica en Enfermedades Infecciosas y Parasitarias, Mexico City, Mexico; [96]Instituto Mexicano del Seguro Social (IMSS), Centro Médico Nacional La Raza, Hospital de Infectología, Mexico City, Mexico; [97]Clinica Comfamiliar Risaralda, Pereira, Colombia; [98]Bellvitge Biomedical Research Institute (IDIBELL), Neurometabolic Diseases Laboratory, L'Hospitalet de Llobregat, Barcelona, Spain; [99]Catalan Institution of Research and Advanced Studies (ICREA), Barcelona, Spain; [100]Hospital Ophir Loyola, Departamento de Ensino e Pesquisa, Belém, Brazil; [101]Unidad de Cuidados Intensivos, Hospital Clínico Universitario de Santiago (CHUS), Sistema Galego de Saúde (SERGAS), Santiago de Compostela, Spain; [102]Department of Preventive Medicine and Public Health, School of Medicine, Universidad Autónoma de Madrid, Madrid, Spain; [103]IdiPaz (Instituto de Investigación Sanitaria Hospital Universitario La Paz), Madrid, Spain; [104]IMDEA-Food Institute, CEI UAM+CSIC, Madrid, Spain; [105]Complejo Asistencial Universitario de León, León, Spain; [106]Instituto de Investigación Biosanitaria de Granada (ibs GRANADA), Granada, Spain; [107]Universidad de Granada, Departamento Bioquímica, Biología Molecular e Inmunología III, Granada, Spain; [108]Hospital Infanta Elena, Allergy Unit, Valdemoro, Madrid, Spain; [109]Instituto de Investigación Sanitaria-Fundación Jiménez Díaz University Hospital - Universidad Autónoma de Madrid (IIS-FJD, UAM), Madrid, Spain; [110]Faculty of Medicine, Universidad Francisco de Vitoria, Madrid, Spain; [111]Hospital Universitario Infanta Leonor, Madrid, Spain; [112]Complutense University of Madrid, Madrid, Spain; [113]Gregorio Marañón Health Research Institute (IiSGM), Madrid, Spain; [114]Haemostasis and Thrombosis Unit, Hospital de la Santa Creu i Sant Pau, IIB Sant Pau, Barcelona, Spain; [115]Hospital Clinico Universitario de Valladolid, Servicio de Anestesiologia y Reanimación, Valladolid, Spain; [116]Universidad de Valladolid, Departamento de Cirugía, Valladolid, Spain; [117]Hospital Clinico Universitario de Valladolid, Servicio de Hematologia y Hemoterapia, Valladolid, Spain; [118]Hospital de Niños Ricardo Gutierrez, Buenos Aires, Argentina; [119]Fundación Universitaria de Ciencias de la Salud, Bogotá, Colombia; [120]Spanish National Cancer Research Centre, Familial Cancer Clinical Unit, Madrid, Spain; [121]University Hospital of Burgos, Burgos, Spain; [122]Universidad Simón Bolívar, Facultad de Ciencias de la Salud, Barranquilla, Colombia; [123]Centro para el Desarrollo de la Investigación Científica, Asunción, Paraguay; [124]Centre for Biomedical Network Research on Neurodegenerative Diseases (CIBERNED), Instituto de Salud Carlos III, Madrid, Spain; [125]Research Center and Memory clinic, ACE Alzheimer Center Barcelona, Universitat Internacional de Catalunya, Barcelona, Spain; [126]CIEN Foundation/Queen Sofia Foundation Alzheimer Center, Madrid, Spain; [127]Hospital Universitario de Valme, Unidad Clínica de Enfermedades Infecciosas y Microbiología, Sevilla, Spain; [128]Sección Genética Médica - Servicio de Pediatría, Hospital Clínico Universitario Virgen de la Arrixaca, Servicio Murciano de Salud, Murcia, Spain; [129]Departamento Cirugía, Pediatría, Obstetricia y Ginecología, Facultad de Medicina, Universidad de Murcia (UMU), Murcia, Spain; [130]Grupo Clínico Vinculado, Centre for Biomedical Network Research on Rare Diseases (CIBERER), Instituto de Salud Carlos III, Madrid, Spain; [131]Department of Anthropology, University of Toronto at Mississauga, Mississauga, Canada; [132]Tecnologico de Monterrey, Escuela de Medicina y Ciencias de la Salud, Monterrey, Mexico; [133]Research Unit, Hospital Universitario Nuestra Señora de Candelaria, Instituto de Investigación Sanitaria de Canarias, Santa Cruz de Tenerife, Spain; [134]Department of Clinical Sciences, University Fernando Pessoa Canarias, Las Palmas de Gran Canaria, Spain; [135]Centre for Biomedical Network Research on Respiratory Diseases (CIBERES), Instituto de Salud Carlos III, Madrid, Spain

**Abstract** The genetic basis of severe COVID-19 has been thoroughly studied, and many genetic risk factors shared between populations have been identified. However, reduced sample sizes from non-European groups have limited the discovery of population-specific common risk loci. In this second study nested in the SCOURGE consortium, we conducted a genome-wide association study (GWAS) for COVID-19 hospitalization in admixed Americans, comprising a total of 4702 hospitalized cases recruited by SCOURGE and seven other participating studies in the COVID-19 Host Genetic Initiative. We identified four genome-wide significant associations, two of which constitute novel loci and were first discovered in Latin American populations (*BAZ2B* and *DDIAS*). A trans-ethnic meta-analysis revealed another novel cross-population risk locus in *CREBBP*. Finally, we assessed the performance of a cross-ancestry polygenic risk score in the SCOURGE admixed American cohort. This study constitutes the largest GWAS for COVID-19 hospitalization in admixed Latin Americans conducted to date. This allowed to reveal novel risk loci and emphasize the need of considering the diversity of populations in genomic research.

## eLife assessment

The authors conducted a **valuable** GWAS meta-analysis for COVID-19 hospitalization in admixed American populations and prioritized risk variants and genes. The evidence supporting the claims of the authors is **solid**. The work will be of interest to scientists studying the genetic basis of COVID pathogenesis.

## Introduction

To date, more than 50 loci associated with COVID-19 susceptibility, hospitalization, and severity have been identified using genome-wide association studies (GWAS) (*Kanai et al., 2023*; *Pairo-Castineira et al., 2023*). The COVID-19 Host Genetics Initiative (HGI) has made significant efforts (*Niemi et al., 2021*) to augment the power to identify disease loci by recruiting individuals from diverse populations and conducting a trans-ancestry meta-analysis. Despite this, the lack of genetic diversity and a focus on cases of European ancestries still predominate in the studies (*Popejoy and Fullerton, 2016*; *Sirugo et al., 2019*). In addition, while trans-ancestry meta-analyses are a powerful approach for discovering shared genetic risk variants with similar effects across populations (*Li and Keating, 2014*), they may fail to identify risk variants that have larger effects on particular underrepresented populations. Genetic disease risk has been shaped by the particular evolutionary history of populations and environmental exposures (*Rosenberg et al., 2010*). Their action is particularly important for infectious diseases due to the selective constraints that are imposed by host–pathogen interactions (*Karlsson et al., 2014*; *Kwok et al., 2021*). Literature examples of this in COVID-19 severity include a *DOCK2* gene variant in East Asians (*Namkoong et al., 2022*) and frequent loss-of-function variants in *IFNAR1* and *IFNAR2* genes in Polynesian and Inuit populations, respectively (*Bastard et al., 2022*; *Duncan et al., 2022*).

Including diverse populations in case–control GWAS with unrelated participants usually requires a prior classification of individuals in genetically homogeneous groups, which are typically analyzed separately to control the population stratification effects (*Peterson et al., 2019*). Populations with recent admixture impose an additional challenge to GWASs due to their complex genetic diversity and linkage disequilibrium (LD) patterns, requiring the development of alternative approaches and a careful inspection of results to reduce false positives due to population structure (*Rosenberg et al., 2010*). In fact, there are benefits in study power from modeling the admixed ancestries either locally, at the regional scale in the chromosomes, or globally, across the genome, depending on factors such as the heterogeneity of the risk variant in frequencies or the effects among the ancestry strata (*Mester et al., 2023*). Despite the development of novel methods specifically tailored for the analysis of admixed populations (*Atkinson et al., 2021*), the lack of a standardized analysis framework and the difficulties in confidently clustering admixed individuals into particular genetic groups often lead to their exclusion from GWAS.

The Spanish Coalition to Unlock Research on Host Genetics on COVID-19 (SCOURGE) recruited COVID-19 patients between March and December 2020 from hospitals across Spain and from March

**Table 1.** Demographic characteristics of the SCOURGE Latin American cohort.

| Variable | | Non-hospitalized (N = 1887) | Hospitalized (N = 1625) |
|---|---|---|---|
| Age, mean years ±SD | | 39.1 ± 11.9 | 54.1 ± 14.5 |
| Sex, N (%) | | | |
| | Female (%) | 1253 (66.4) | 668 (41.1) |
| Global genetic inferred ancestry, % mean ± SD | | | |
| | European | 54.4 ± 16.2 | 39.4 ± 20.7 |
| | African | 15.3 ± 12.7 | 9.1 ± 11.6 |
| | Native American | 30.3 ± 19.8 | 51.3 ± 26.5 |
| Comorbidities, N (%) | | | |
| | Vascular/endocrinological | 488 (25.9) | 888 (64.5) |
| | Cardiac | 60 (3.2) | 151 (9.3) |
| | Nervous | 15 (0.8) | 61 (3.8) |
| | Digestive | 14 (0.7) | 33 (2.0) |
| | Onco-hematological | 21 (1.1) | 48 (3.00) |
| | Respiratory | 76 (4.0) | 118 (7.3) |

2020 to July 2021 in Latin America (https://www.scourge-covid.org). A first GWAS of COVID-19 severity among Spanish patients of European descent revealed novel disease loci and explored age- and sex-varying effects of genetic factors (*Cruz et al., 2022*). Here, we present the findings of a GWAS meta-analysis in admixed Latin American (AMR) populations, comprising individuals from the SCOURGE Latin American cohort and the HGI studies, which allowed us to identify two novel severe COVID-19 loci, *BAZ2B* and *DDIAS*. Further analyses modeling the admixture from three genetic ancestral components and performing a trans-ethnic meta-analysis led to the identification of an additional risk locus near *CREBBP*. We finally assessed a cross-ancestry polygenic risk score (PGS) model with variants associated with critical COVID-19.

## Results
### Meta-analysis of COVID-19 hospitalization in admixed Americans
#### Study cohorts
Within the SCOURGE consortium, we included 1608 hospitalized cases and 1887 controls (not hospitalized COVID-19 patients) from Latin American countries and from recruitments of individuals of Latin American descent conducted in Spain (*Supplementary file 1*). Quality control details and estimation of global genetic inferred ancestry (GIA) (*Figure 1—figure supplement 1*) are described in 'Materials and methods', whereas clinical and demographic characteristics of patients included in the analysis are shown in *Table 1*. Summary statistics from the SCOURGE cohort were obtained under a logistic mixed model with the SAIGE model ('Materials and methods'). Another seven studies participating in the COVID-19 HGI consortium were included in the meta-analysis of COVID-19 hospitalization in admixed Americans (*Figure 1*).

#### GWAS meta-analysis
We performed a fixed-effects GWAS meta-analysis using the inverse of the variance as weights for the overlapping markers. The combined GWAS sample size consisted of 4702 admixed AMR hospitalized cases and 68,573 controls.

This GWAS meta-analysis revealed genome-wide significant associations at four risk loci (*Table 2*, *Figure 2*; a quantile–quantile plot is shown in *Figure 2—figure supplement 1*), two of which (*BAZ2B* and *DDIAS*) were novel discoveries. A Meta-Analysis Model-based Assessment of replicability

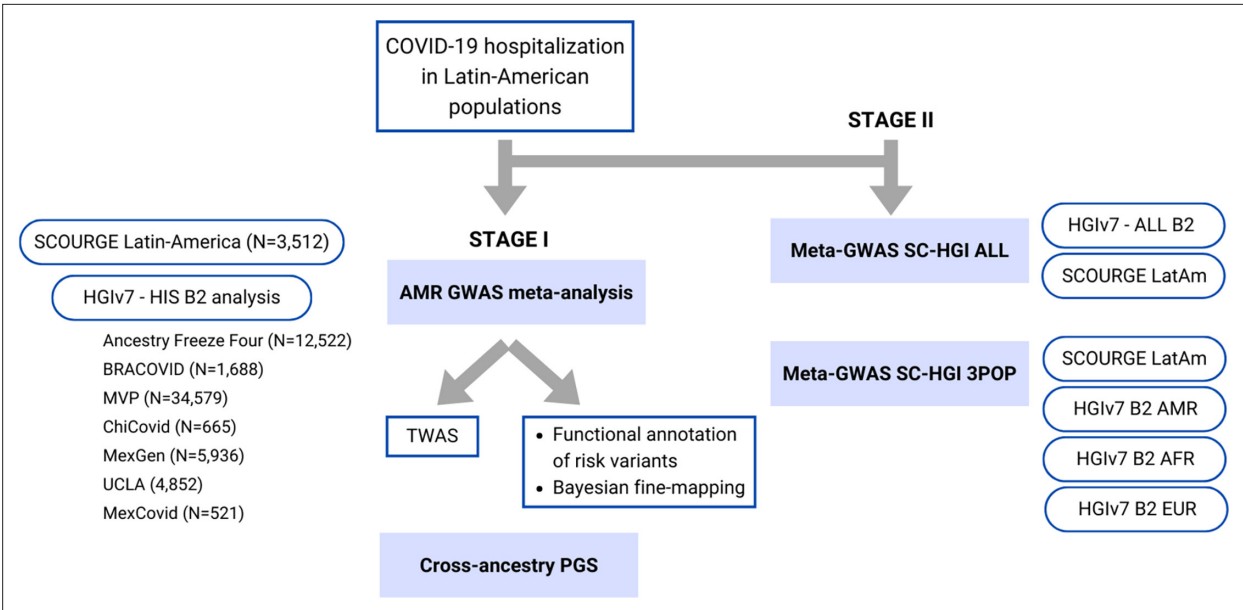

**Figure 1.** Flow chart of this study. Stage I of the study involved a meta-analysis of the Latin American genome-wide association studies (GWAS) from SCOURGE and the COVID-19 Host Genetics Initiative. The resulting meta-analysis was leveraged to prioritize genes by using a transcriptome-wide association study (TWAS), Bayesian fine-mapping and functional annotations, and to assess the generalizability of polygenic risk score (PGS) cross-population models in Latin Americans. Stage II involved two additional cross-population GWAS meta-analyses to further investigate the replicability of findings.

The online version of this article includes the following figure supplement(s) for figure 1:

**Figure supplement 1.** Global genetic inferred ancestry (GIA) composition in the SCOURGE Latin American cohort.

(MAMBA) approach to leverage the strength and consistency of associations across the contributing studies supported >90% likelihood for one of the novel loci to likely replicate in future studies (*Table 3*). Four lead variants were identified, linked to other 310 variants (*Supplementary files 2 and 3*). A gene-based association test revealed a significant association in *BAZ2B* and in previously known COVID-19 risk loci: *LZTFL1*, *XCR1*, *FYCO1*, *CCR9*, and *IFNAR2* (*Supplementary file 4*).

Located within the *BAZ2B* gene, the sentinel variant rs13003835 (*Figure 3*) is an intronic variant associated with an increased risk of COVID-19 hospitalization (odds ratio [OR]=1.20, 95% confidence interval [CI] = 1.12–1.27, p=3.66 × 10⁻⁸). This association was not previously reported in any GWAS of COVID-19 published to date. Interestingly, rs13003835 did not reach significance (p=0.972) in the COVID-19 HGI trans-ancestry meta-analysis including the five population groups (*Kanai et al., 2023*).

The other novel risk locus is led by the sentinel variant rs77599934 (*Figure 3*), a rare intronic variant located in chromosome 11 within *DDIAS* and associated with the risk of COVID-19 hospitalization (OR = 2.27, 95% CI = 1.70–3.04, p=2.26 × 10⁻⁸).

We also observed a suggestive association with rs2601183 in chromosome 15, which is located between *ZNF774* and *IQGAP1* (allele-G OR = 1.20, 95% CI = 1.12–1.29, p=6.11 × 10⁻⁸, see *Supplementary file 2*), which has not yet been reported in any other GWAS of COVID-19 to date.

**Table 2.** Lead independent variants in the admixed AMR genome-wide association studies (GWAS) meta-analysis.

| SNP rsID | chr:pos | EA | NEA | OR (95% CI) | p-Value | EAF cases | EAF controls | Nearest gene | Mamba PPR |
|----------|---------|----|----|-----------|---------|-----------|--------------|--------------|-----------|
| *rs13003835* | 2:159407982 | T | C | 1.20 (1.12–1.27) | 3.66E-08 | 0.563 | 0.429 | *BAZ2B* | 0.30 |
| *rs35731912* | 3:45848457 | T | C | 1.65 (1.47–1.85) | 6.30E-17 | 0.087 | 0.056 | *LZTFL1* | 0.95 |
| *rs2477820* | 6:41535254 | A | T | 0.84 (0.79–0.89) | 1.89E-08 | 0.453 | 0.517 | *FOXP4-AS1* | 0.18 |
| *rs77599934* | 11:82906875 | G | A | 2.27 (1.7–3.04) | 2.26E-08 | 0.016 | 0.011 | *DDIAS* | 0.95 |

EA: effect allele. NEA: noneffect allele. EAF: effect allele frequency in the SCOURGE study. PPR: posterior probability of replicability.

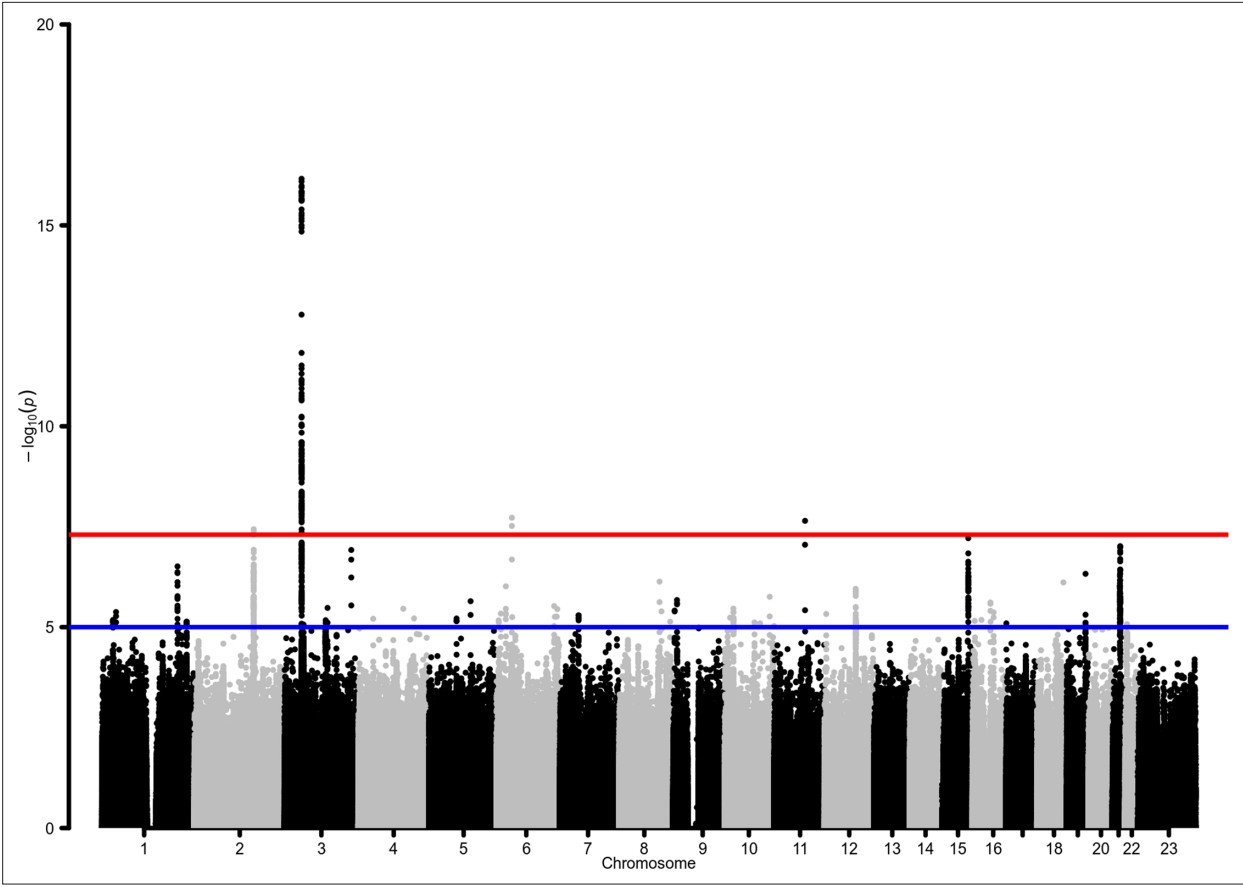

**Figure 2.** Manhattan plot for the admixed AMR genome-wide association studies (GWAS) meta-analysis. Probability thresholds at $p=5 \times 10^{-8}$ and $p=5 \times 10^{-5}$ are indicated by the horizontal lines. Genome-wide significant associations with COVID-19 hospitalizations were found on chromosome 2 (within *BAZ2B*), chromosome 3 (within *LZTFL1*), chromosome 6 (within *FOXP4*), and chromosome 11 (within *DDIAS*).

The online version of this article includes the following figure supplement(s) for figure 2:

**Figure supplement 1.** Quantile–quantile plot for the AMR genome-wide association studies (GWAS) meta-analysis.

The GWAS meta-analysis also pinpointed two significant variants at known loci, *LZTFL1* and *FOXP4*. The SNP rs35731912 was previously associated with COVID-19 severity in EUR populations (***Degenhardt et al., 2022***), and it was mapped to *LZTFL1*. While rs2477820 is a novel risk variant within the *FOXP4 gene*, it has a moderate LD ($r^2 = 0.295$) with rs2496644, which has been linked to COVID-19 hospitalization (***Kousathanas et al., 2022***). This is consistent with the effects of LD in tag-SNPs when conducting GWAS in diverse populations.

None of the lead variants was associated with the comorbidities included in *Table 1*.

**Table 3.** Novel variants in the SC-HGI$_{ALL}$ and SC-HGI$_{3POP}$ meta-analyses (with respect to HGIv7). Independent signals after LD clumping.

| SNP rsID | chr:pos | EA | NEA | OR (95% CI) | p-Value | Nearest gene | Analysis |
|---|---|---|---|---|---|---|---|
| *rs76564172* | 16:3892266 | T | G | 1.31 (1.19–1.44) | 9.64E-09 | *CREBBP* | SC-HGI$_{3POP}$ |
| *rs66833742* | 19:4063488 | T | C | 0.94 (0.92–0.96) | 1.89E-08 | *ZBTB7A* | SC-HGI$_{3POP}$ |
| *rs66833742* | 19:4063488 | T | C | 0.94 (0.92–0.96) | 2.50E-08 | *ZBTB7A* | SC-HGI$_{ALL}$ |
| *rs2876034* | 20:6492834 | A | T | 0.95 (0.93–0.97) | 2.83E-08 | *CASC20* | SC-HGI$_{ALL}$ |

EA: effect allele. NEA: non-effect allele.

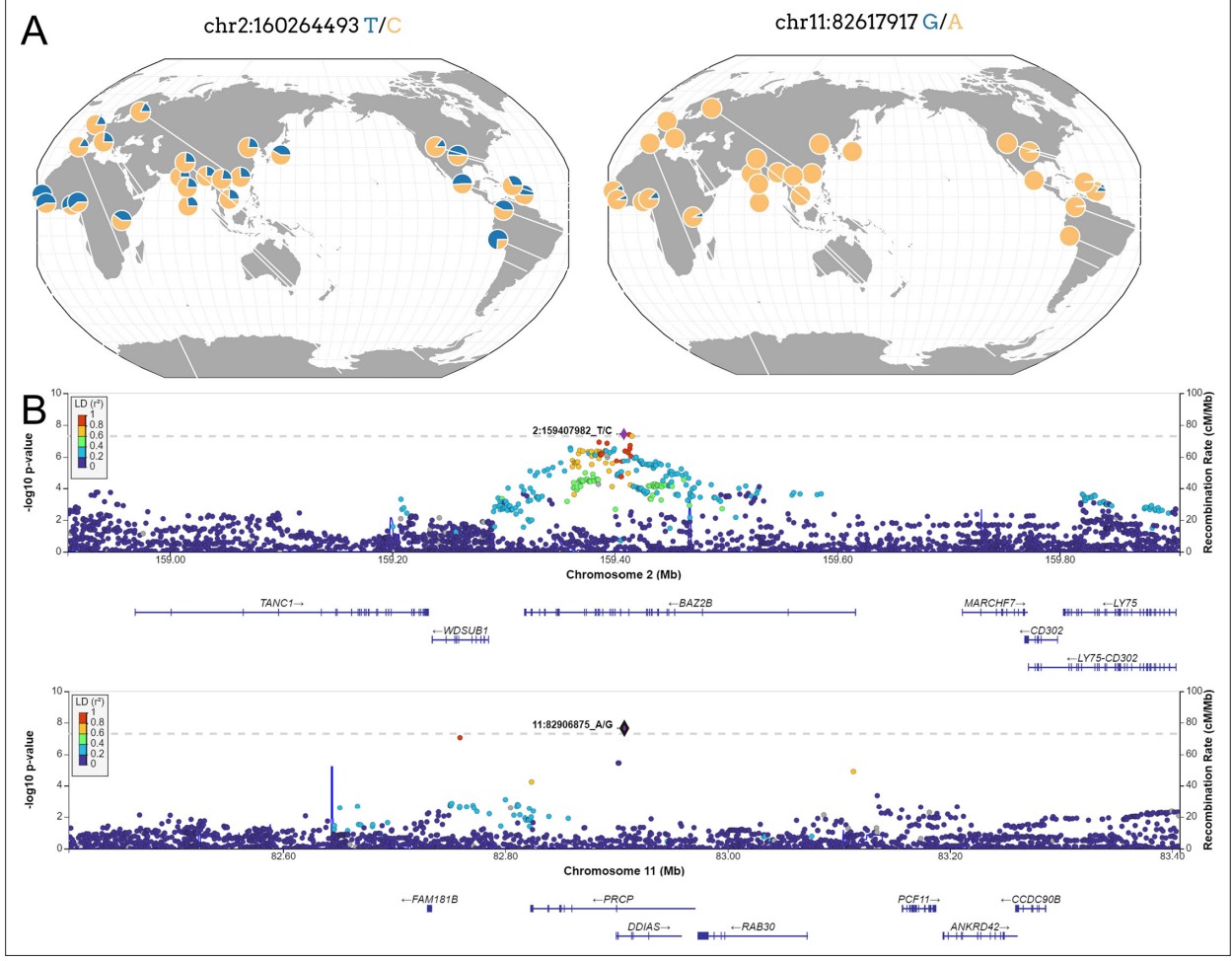

**Figure 3.** New loci associated with COVID-19 hospitalization in Admixed american populations. (**A**) Regional association plots for rs1003835 at chromosome 2 and rs77599934 at chromosome 11. (**B**) Allele frequency distribution across the 1000 Genomes Project populations for the lead variants rs1003835 and rs77599934. Retrieved from *The Geography of Genetic Variants Web* or GGV.

The online version of this article includes the following figure supplement(s) for figure 3:

**Figure supplement 1.** Regional association plots for the fine mapped loci in chromosomes 2 (**A**) and 16 (**B**).

## Functional mapping of novel risk variants

Variants belonging to the lead loci were prioritized by positional and expression quantitative trait loci (eQTL) mapping with FUMA, resulting in 31 mapped genes (*Supplementary file 5*). Within the region surrounding the lead variant in chromosome 2, FUMA prioritized four genes in addition to *BAZ2B* (*PLA2R1*, *LY75*, *WDSUB1*, and *CD302*). rs13003835 (allele C) is an eQTL of *LY75* in the esophagus mucosa (NES = 0.27) and of *BAZ2B-AS* in whole blood (NES = 0.27), while rs2884110 ($R^2$ = 0.85) is an eQTL of *LY75* in lung (NES = 0.22). As for the chromosome 11, rs77599934 (allele G) is in moderate-to-strong LD ($r^2$ = 0.776) with rs60606421 (G deletion, allele -), which is an eQTL associated with a reduced expression of *DDIAS* in the lungs (NES = −0.49, allele -). The sentinel variant for the region in chromosome 16 is in perfect LD ($r^2$ = 1) with rs601183, an eQTL of *ZNF774* in the lung.

### Bayesian fine mapping

We performed different approaches to narrow down the prioritized loci to a set of most likely genes driving the associations. First, we computed credible sets at the 95% confidence level for causal variants and annotated them with VEP and V2G aggregate scoring. The 95% confidence credible set from the region of chromosome 2 around rs13003835 included 76 variants, which can be found in *Supplementary file 6* and a regional plot is shown in *Figure 3—figure supplement 1* (VEP and V2G

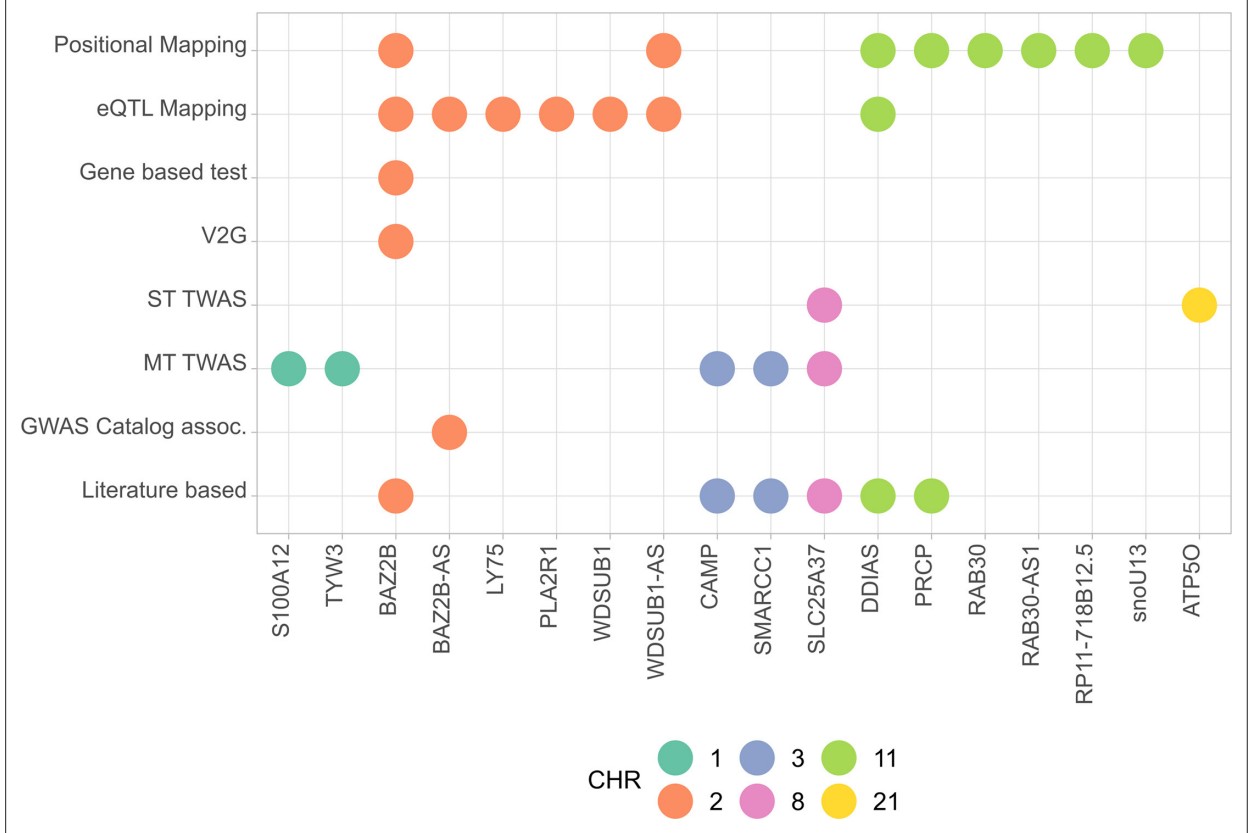

**Figure 4.** Summary of the results from gene prioritization strategies used for genetic associations in AMR populations. Genome-wide association studies (GWAS) catalog association for *BAZ2B-AS* was with FEV/FCV ratio. Literature-based evidence is further explored in 'Discussion'.

The online version of this article includes the following figure supplement(s) for figure 4:

**Figure supplement 1.** Gene−tissue pairs for which either rs1003835 or rs60606421 are significant expression quantitative trait loci (eQTL) at false discovery rate (FDR) < 0.05 (data retrieved from https://gtexportal.org/home/snp/).

annotations are included in *Supplementary files 7 and 8*). The V2G score prioritized *BAZ2B* as the most likely gene driving the association. However, the approach was unable to converge allocating variants in a 95% confidence credible set for the region in chromosome 11.

## Transcriptome-wide association study (TWAS)

Five novel genes, namely, *SLC25A37*, *SMARCC1*, *CAMP*, *TYW3*, and *S100A12* (*Supplementary file 9*), were found to be significantly associated in the cross-tissue TWAS. To our knowledge, these genes have not been reported previously in any COVID-19 TWAS or GWAS analyses published to date. In the single-tissue analyses, *ATP5O* and *CXCR6* were significantly associated in the lungs, *CCR9* was significantly associated in whole blood, and *IFNAR2* and *SLC25A37* were associated in lymphocytes.

Likewise, we carried out TWAS analyses using the models trained in the admixed populations. However, no significant gene pairs were detected in this case. The top 10 genes with the lowest p-values for each of the datasets (Puerto Ricans, Mexicans, African Americans, and pooled cohorts) are shown in *Supplementary file 10*. Although not significant, *KCNC3* was repeated in the four analyses, whereas *MAPKAPK3*, *NAPSA*, and *THAP5* were repeated in three out of four. Both *NAPSA* and *KCNC3* are located in the chromosome 19 and were reported in the latest HGI meta-analysis (*Kanai et al., 2023*).

All mapped genes from analyses conducted in AMR populations are shown in *Figure 4*, and associations for the two novel variants with expression are shown in *Figure 4—figure supplement 1*.

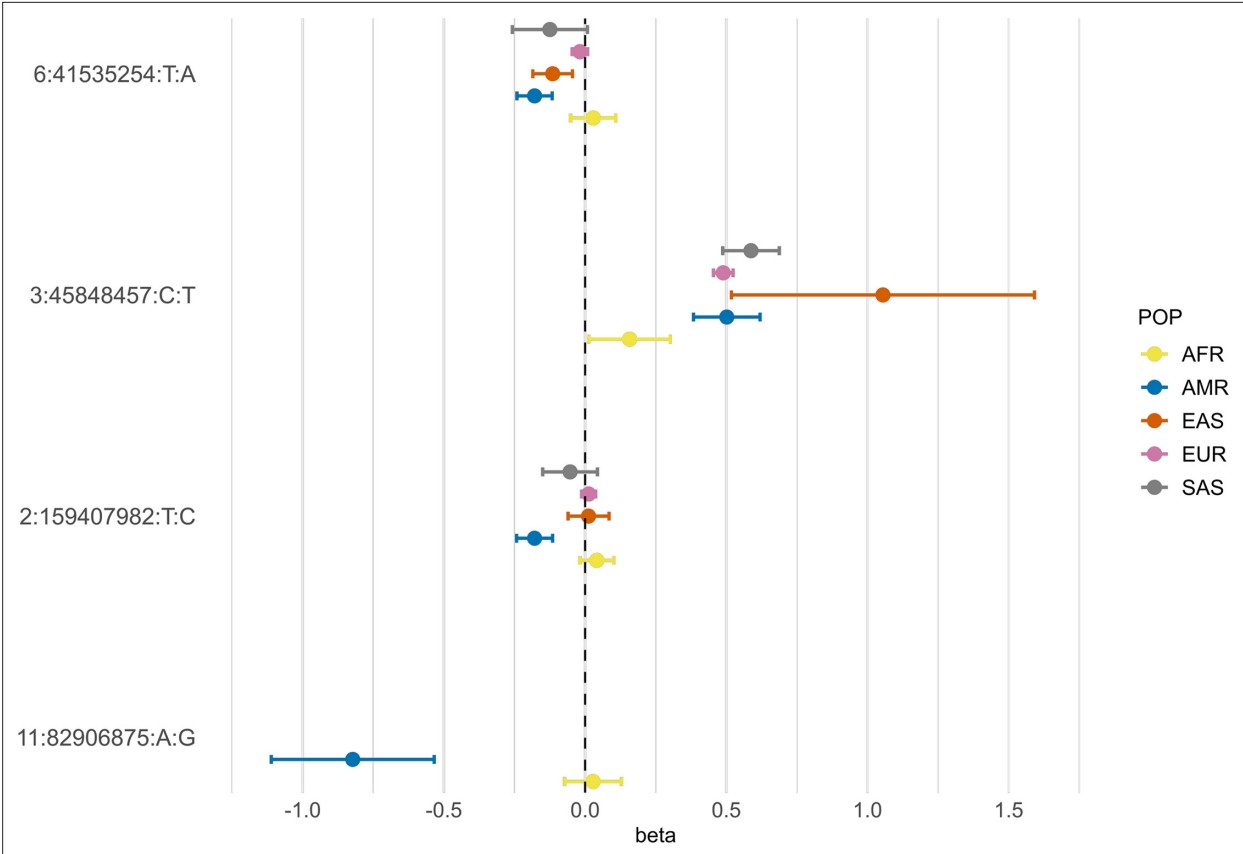

**Figure 5.** Forest plot showing effect sizes and the corresponding confidence intervals for the sentinel variants identified in the AMR meta-analysis across populations. All beta values with their corresponding CIs were retrieved from the B2 population-specific meta-analysis from the HGI v7 release, except for AMR, for which the beta value and IC from the HGI$_{AMR}$-SCOURGE meta-analysis are represented.

## Genetic architecture of COVID-19 hospitalization in AMR populations

### Allele frequencies of rs13003835 and rs77599934 across ancestries

Neither rs13003835 (*BAZ2B*) nor rs77599934 (*DDIAS*) were significantly associated in the COVID-19 HGI B2 cross-population or population-specific meta-analyses. Thus, we investigated their allele frequencies (AF) across populations and compared their effect sizes.

According to gnomAD v3.1.2, the T allele at rs13003835 (*BAZ2B*) has an AF of 43% in admixed AMR groups, while AF is lower in the EUR populations (16%) and in the global sample (29%). Local ancestry inference (LAI) reported by gnomAD shows that within the Native American component, the risk allele T is the major allele, whereas it is the minor allele within the African and European LAI components. These large differences in AF might be the reason underlying the association found in AMR populations. However, when comparing effect sizes between populations, we found that they were in opposite directions between SAS-AMR and EUR-AFR-EAS and that there was large heterogeneity among them (*Figure 5*). We queried SNPs within 50 kb windows of the lead variant in each of the other populations that had p-value <0.01. The variant with the lowest p-value in the EUR population was rs559179177 (p=1.72 × 10⁻⁴), which is in perfect LD (r² = 1) in the 1KGP EUR population with our sentinel variant (rs13003835), and in moderate LD r² = 0.4 in AMR populations. Since this variant was absent from the AMR analysis, probably due to its low frequency, it could not be meta-analyzed. Power calculations revealed that the EUR analysis was underpowered for this variant to achieve genome-wide significance (77.6%, assuming an effect size of 0.46, EAF = 0.0027, and number of cases/controls as shown in the HGI website for B2-EUR). In the cross-population meta-analysis (B2-ALL), rs559179177 obtained a p-value of 5.9 × 10⁻⁴.

rs77599934 (*DDIAS*) had an AF of 1.1% for the G allele in the non-hospitalized controls (*Table 2*), in line with the recorded gnomAD AF of 1% in admixed AMR groups. This variant has the potential

to be a population-specific variant, given the allele frequencies in other population groups, such as EUR (0% in Finnish, 0.025% in non-Finnish), EAS (0%) and SAS (0.042%), and its greater effect size over AFR populations (*Figure 5*). Examining the LAI, the G allele occurs at a 10.8% frequency in the African component, while it is almost absent in the Native American and European. Due to its low MAF, rs77599934 was not analyzed in the COVID-19 HGI B2 cross-population meta-analysis and was only present in the HGI B2 AFR population-specific meta-analysis, precluding the comparison (*Figure 5*). For this reason, we retrieved the variant with the lowest p-value within a 50 kb region around rs77599934 in the COVID-19 HGI cross-population analysis to investigate whether it was in moderate-to-strong LD with our sentinel variant. The variant with the smallest p-value was rs75684040 (OR = 1.07, 95% CI = 1.03–1.12, p=1.84 × $10^{-3}$). However, LD calculations using the 1KGP phase 3 dataset indicated that rs77599934 and rs75684040 were poorly correlated ($r^2$ = 0.11). As for AFR populations, the variant with the lowest p-value was rs138860115 (p=8.3 × $10^{-3}$), but it was not correlated with the lead SNP of this locus.

## Cross-population meta-analyses

We carried out two cross-ancestry inverse variance-weighted fixed-effects meta-analyses with the admixed AMR GWAS meta-analysis results to evaluate whether the discovered risk loci replicated when considering other population groups. In doing so, we also identified novel cross-population COVID-19 hospitalization risk loci.

First, we combined the SCOURGE Latin American GWAS results with the HGI B2 ALL analysis (*Supplementary file 11*). We refer to this analysis as the SC-HGI$_{ALL}$ meta-analysis. Out of the 40 genome-wide significant loci associated with COVID-19 hospitalization in the last HGI release (*Kanai et al., 2023*), this study replicated 39, and the association was stronger than in the original study in 29 of those (*Supplementary file 12*). However, the variant rs13003835 located in *BAZ2B* did not replicate (OR = 1.00, 95% CI = 0.98–1.03, p=0.644).

In this cross-ancestry meta-analysis, we replicated two associations that were not found in HGIv7, albeit they were sentinel variants in the latest GenOMICC meta-analysis (*Pairo-Castineira et al., 2023*). We found an association at the *CASC20* locus led by the variant rs2876034 (OR = 0.95, 95% CI = 0.93–0.97, p=2.83 × $10^{-8}$). This variant is in strong LD with the sentinel variant of that study (rs2326788, $r^2$ = 0.92), which was associated with critical COVID-19 (*Pairo-Castineira et al., 2023*). In addition, this meta-analysis identified the variant rs66833742 near *ZBTB7A* associated with COVID-19 hospitalization (OR = 0.94, 95% CI = 0.92–0.96, p=2.50 × $10^{-8}$). Notably, rs66833742 or its perfect proxy rs67602344 ($r^2$ = 1) are also associated with upregulation of *ZBTB7A* in whole blood and in esophageal mucosa. This variant was previously associated with COVID-19 hospitalization (*Pairo-Castineira et al., 2023*).

In a second analysis, we also explored the associations across the defined admixed AMR, EUR, and AFR ancestral sources by combining through meta-analysis the SCOURGE Latin American GWAS results with the HGI studies in EUR, AFR, and admixed AMR and excluding those from EAS and SAS (*Supplementary file 13*). We refer to this as the SC-HGI$_{3POP}$ meta-analysis. The association at rs13003835 (*BAZ2B*, OR = 1.01, 95% CI = 0.98–1.03, p=0.605) was not replicated, and rs77599934 near *DDIAS* could not be assessed, although the association at the *ZBTB7A* locus was confirmed (rs66833742, OR = 0.94, 95% CI = 0.92–0.96, p=1.89 × $10^{-8}$). The variant rs76564172 located near *CREBBP* also reached statistical significance (OR = 1.31, 95% CI = 1.25–1.38, p=9.64 × $10^{-9}$). The sentinel variant of the region linked to *CREBBP* (in the trans-ancestry meta-analysis) was also subjected to a Bayesian fine mapping (*Supplementary file 6*). Eight variants were included in the credible set for the region in chromosome 16 (meta-analysis SC-HGI$_{3POP}$).

## Polygenic risk score models

Using the 49 variants associated with disease severity that are shared across populations according to the HGIv7, we constructed a PGS model to assess its generalizability in the admixed AMR (*Supplementary file 14*). First, we calculated the PGS for the SCOURGE Latin Americans and explored the association with COVID-19 hospitalization under a logistic regression model. The PGS model was associated with a 1.48-fold increase in COVID-19 hospitalization risk per every PGS standard deviation. It also contributed to explaining a slightly larger variance ($\Delta R^2$ = 1.07%) than the baseline model.

Subsequently, we divided the individuals into PGS deciles and percentiles to assess their risk stratification. The median percentile among controls was 40, while in cases, it was 63. Those in the top PGS decile exhibited a 2.89-fold (95% CI = 2.37–3.54, p=1.29 × $10^{-7}$) greater risk compared to individuals in the deciles between 4 and 6 (corresponding to a score of the median distribution).

We also examined the distribution of PGS across a five-level severity scale to further determine if there was any correspondence between clinical severity and genetic risk. Median PGS were lower in the asymptomatic and mild groups, whereas higher median scores were observed in the moderate, severe, and critical patients (*Figure 6*). We fitted a multinomial model using the asymptomatic class as a reference and calculated the OR for each category (*Supplementary file 15*), observing that the disease genetic risk was similar among asymptomatic, mild, and moderate patients. Given that the PGS was built with variants associated with critical disease and/or hospitalization and that the categories severe and critical correspond to hospitalized patients, these results underscore the ability of cross-ancestry PGS for risk stratification even in an admixed population.

## Discussion

We have conducted the largest GWAS meta-analysis of COVID-19 hospitalization in admixed AMR to date. While the genetic risk basis discovered for COVID-19 is largely shared among populations, trans-ancestry meta-analyses on this disease have primarily included EUR samples. This dominance of GWAS in Europeans and the subsequent bias in sample sizes can mask population-specific genetic risks (i.e., variants that are monomorphic in some populations) or be less powered to detect risk variants having higher allele frequencies in population groups other than Europeans. In this sense, after combining data from admixed AMR patients, we found two risk loci that were first discovered in a GWAS of Latin American populations. Interestingly, the sentinel variant rs77599934 in the *DDIAS* gene is a rare coding variant (~1% for allele G) with a large effect on COVID-19 hospitalization that is nearly monomorphic in most of the other populations. This has likely led to its exclusion from the cross-population meta-analyses conducted to date, remaining undetectable.

Fine mapping of the region harboring *DDIAS* did not reveal further information about which gene could be the more prone to be causal or about the functional consequences of the risk variant, but our sentinel variant was in strong LD with an eQTL that associated with reduced gene expression of *DDIAS* in the lung. *DDIAS,* known as damage-induced apoptosis suppressor gene, is itself a plausible candidate gene. It has been linked to DNA damage repair mechanisms: research has shown that depletion of *DDIAS* leads to an increase in ATM phosphorylation and the formation of p53-binding protein (53BP1) foci, a known biomarker of DNA double-strand breaks, suggesting a potential role in double-strand break repair (*Brunette et al., 2019*). Interestingly, a study found that infection by SARS-CoV-2 also triggered the phosphorylation of ATM kinase and inhibited repair mechanisms, causing the accumulation of DNA damage (*Gioia et al., 2023*). This gene has also been proposed as a potential biomarker for lung cancer after finding that it interacts with STAT3 in lung cancer cells, regulating IL-6 (*Im et al., 2020*; *Im et al., 2023*) and thus mediating inflammatory processes, while another study determined that its blockade inhibited lung cancer cell growth (*Won et al., 2014*). Another prioritized gene from this region was *PRCP*, an angiotensinase that shares substrate specificity with ACE2 receptor. It has been positively linked to hypertension and some studies have raised hypotheses on its role in COVID-19 progression, particularly in relation to the development of pro-thrombotic events (*Angeli et al., 2023*; *Silva-Aguiar et al., 2020*).

The risk region found in chromosome 2 harbors more than one gene. The lead variant rs13003835 is located within *BAZ2B*, and it increases the expression of the antisense *BAZ2B* gene in whole blood. *BAZ2B* encodes one of the regulatory subunits of the Imitation switch (ISWI) chromatin remodelers (*Li et al., 2021*) constituting the BRF-1/BRF-5 complexes with SMARCA1 and SMARCA5, respectively. Interestingly, it was discovered that *lnc-BAZ2B* promotes macrophage activation through regulation of *BAZ2B* expression. Its overexpression resulted in pulmonary inflammation and elevated levels of *MUC5AC* in mice with asthma (*Xia et al., 2021*). This variant was also an eQTL for *LY75* (encoding lymphocyte antigen 75) in the esophageal mucosa tissue. Lymphocyte antigen 75 is involved in immune processes through antigen presentation in dendritic cells and endocytosis (*Mahnke et al., 2000*) and has been associated with inflammatory diseases, representing a compelling candidate for the region. Increased expression of *LY75* has been detected within hours after infection by SARS-CoV-2 (*Mitchell et al., 2013*; *Sims et al., 2013*). It is worth noting that differences in AF for this variant suggest that

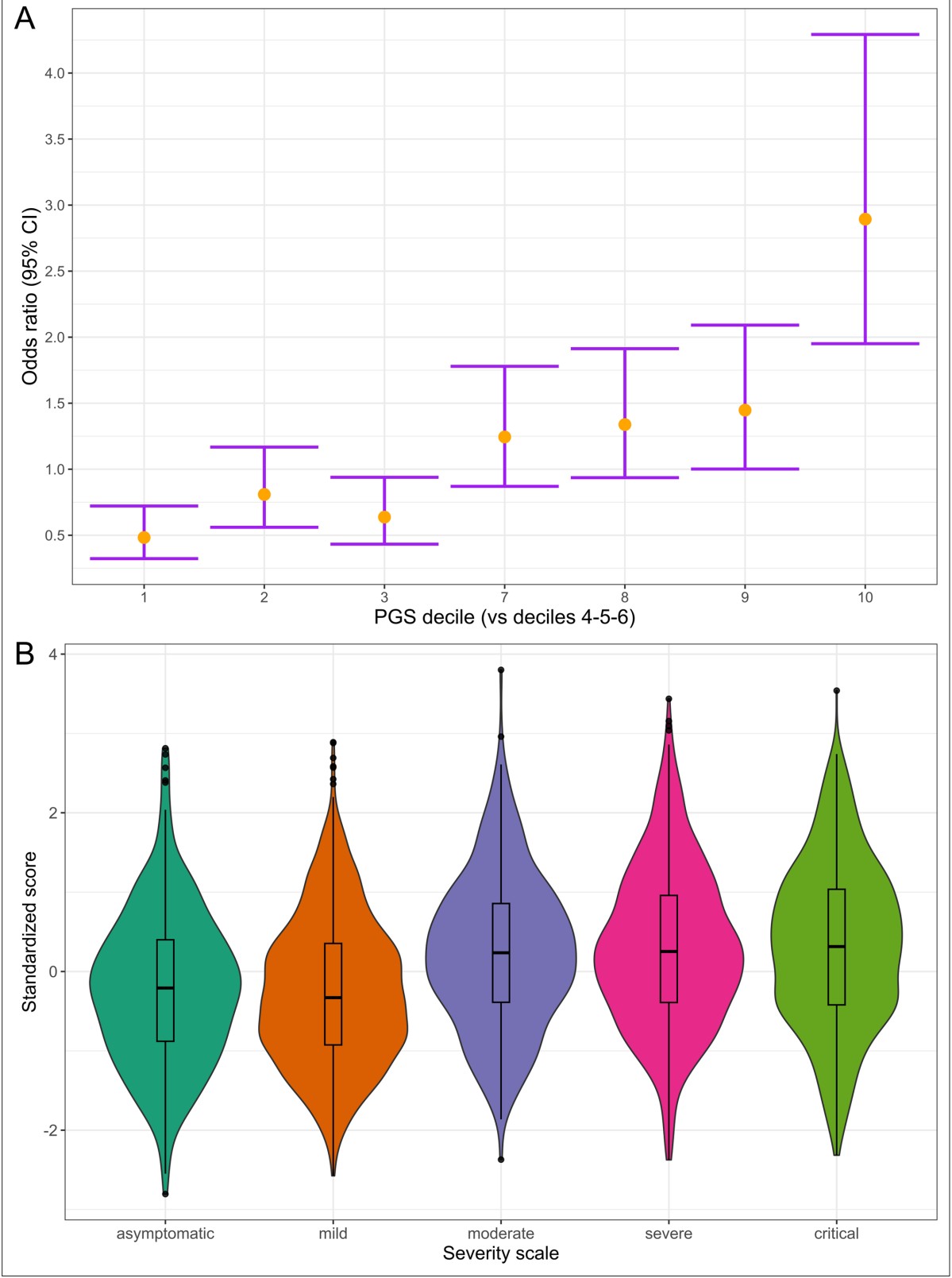

**Figure 6.** Polygenic risk distribution for COVID-19 hospitalization. (**A**) Polygenic risk stratified by polygenic risk score (PGS) deciles comparing each risk group against the lowest risk group (OR–95% CI). (**B**) Distribution of the PGS in each of the severity scale classes. 0, asymptomatic; 1, mild disease; 2, moderate disease; 3, severe disease; 4, critical disease.

analyses in AMR populations might be more powered to detect the association, supporting the necessity of population-specific studies.

A third novel risk region was observed on chromosome 15 between the *IQGAP1* and *ZNF774* genes, although it did not reach genome-wide significance.

Secondary analyses revealed five TWAS-associated genes, some of which have already been linked to severe COVID-19. In a comprehensive multitissue gene expression profiling study (*Gómez-Carballa et al., 2022*), decreased expression of *CAMP* and *S100A8/S100A9* genes in patients with severe COVID-19 was observed, while another study detected the upregulation of *SCL25A37* among patients with severe COVID-19 (*Policard et al., 2021*). *SMARCC1* is a subunit of the SWI/SNF chromatin remodeling complex that has been identified as proviral for SARS-CoV-2 and other coronavirus strains through a genome-wide screen (*Wei et al., 2021*). This complex is crucial for *ACE2* expression and viral entry into the cell (*Wei et al., 2023*). However, it should be noted that using eQTL mostly from European populations such as those in GTEx could result in reduced power to detect associations.

To explore the genetic architecture of the trait among admixed AMR populations, we performed two cross-ancestry meta-analyses including the SCOURGE Latin American cohort GWAS findings. We found that the two novel risk variants were not associated with COVID-19 hospitalization outside the population-specific meta-analysis, highlighting the importance of complementing trans-ancestry meta-analyses with group-specific analyses. Notably, this analysis did not replicate the association at the *DSTYK* locus, which was associated with severe COVID-19 in Brazilian individuals with higher European admixture (*Pereira et al., 2022*). This lack of replication aligns with the initial hypothesis of that study suggesting that the risk haplotype was derived from European populations, as we reduced the weight of this ancestral contribution in our study by excluding those individuals.

Moreover, these cross-ancestry meta-analyses pointed to three loci that were not genome-wide significant in the HGIv7 ALL meta-analysis: a novel locus at *CREBBP* and two loci at *ZBTB7A* and *CASC20* that were reported in another meta-analysis. *CREBBP* and *ZBTB7A* achieved a stronger significance when considering only the EUR, AFR, and admixed AMR GIA groups. According to a recent study, elevated levels of the *ZBTB7A* gene promote a quasihomeostatic state between coronaviruses and host cells, preventing cell death by regulating oxidative stress pathways (*Zhu et al., 2022*). This gene is involved in several signaling pathways, such as B- and T-cell differentiation (*Gupta et al., 2020*). On a separate note, *CREBBP* encodes the CREB binding protein (CBP), which is involved in transcriptional activation and is known to positively regulate the type I interferon response through virus-induced phosphorylation of IRF-3 (*Yoneyama et al., 1998*). In addition, the CREBP/CBP interaction has been implicated in SARS-CoV-2 infection (*Yang et al., 2023*) via the cAMP/PKA pathway. In fact, cells with suppressed *CREBBP* gene expression exhibit reduced replication of the so-called Delta and Omicron SARS-CoV-2 variants (*Yang et al., 2023*).

We developed a cross-population PGS model, which effectively stratified individuals based on their genetic risk and demonstrated consistency with the clinical severity classification of the patients. Only a few polygenic scores were derived from COVID-19 GWAS data. *Horowitz et al., 2022* developed a score using 6 and 12 associated variants (PGS ID: PGP000302) and reported an associated OR (top 10% vs rest) of 1.38 for risk of hospitalization in European populations, whereas the OR for Latin American populations was 1.56. Since their sample size and the number of variants included in the PGS were lower, direct comparisons are not straightforward. Nevertheless, our analysis provides the first results for a PRS applied to a relatively large AMR cohort, being of value for future analyses regarding PRS transferability.

This study is subject to limitations, mostly concerning sample recruitment and composition. The SCOURGE Latino American sample size is small, and the GWAS is likely underpowered. Another limitation is the difference in case–control recruitment across sampling regions that, yet controlled for, may reduce the ability to observe significant associations driven by different compositions of the populations. In this sense, the identified risk loci might not replicate in a cohort lacking any of the parental population sources from the three-way admixture. Likewise, we could not explicitly control for socioenvironmental factors that could have affected COVID-19 spread and hospitalization rates, although genetic principal components are known to capture nongenetic factors. Finally, we must acknowledge the lack of a replication cohort. We used all the available GWAS data for COVID-19 hospitalization in admixed AMR in this meta-analysis due to the low number of studies conducted. Therefore, we had no studies to replicate or validate the results. These concerns may be addressed in

the future by including more AMR GWAS in the meta-analysis, both by involving diverse populations in study designs and supporting research from countries in Latin America.

This study provides novel insights into the genetic basis of COVID-19 severity, emphasizing the importance of considering host genetic factors by using non-European populations, especially of admixed sources. Such complementary efforts can pin down new variants and increase our knowledge on the host genetic factors of severe COVID-19.

# Materials and methods

## Key resources table

| Reagent type (species) or resource | Designation | Source or reference | Identifiers | Additional information |
|---|---|---|---|---|
| Commercial assay or kit | Chemagic DNA Blood 100 kit | PerkinElmer Chemagen Technologies GmbH | | |
| Software, algorithm | Axiom Analysis Suite | Thermo Fisher Scientific | | Version 4.0.3.3 |
| Software, algorithm | PLINK | *Purcell et al., 2007*; https://www.cog-genomics.org/plink/ | RRID:SCR_001757 | Version 1.9; v2 |
| Software, algorithm | TOPMed Imputation Server | https://imputation.biodatacatalyst.nhlbi.nih.gov/ | | Version 2 |
| Software, algorithm | ADMIXTURE | *Alexander et al., 2009*; https://dalexander.github.io/admixture/ | RRID:SCR_001263 | Version 1.3.0 |
| Software, algorithm | SAIGEgds | *Zheng and Davis, 2021*; https://www.bioconductor.org/packages/release/bioc/html/SAIGEgds.html | | Version 1.10.0 |
| Software, algorithm | METAL | *Willer et al., 2010*; https://csg.sph.umich.edu/abecasis/metal/ | RRID:SCR_002013 | Version 2011-03-25 |
| Software, algorithm | FUMA | *Watanabe et al., 2017*; https://fuma.ctglab.nl/ | RRID:SCR_017521 | Version 1.5.2 |
| Software, algorithm | MAMBA | *McGuire et al., 2021*; https://github.com/dan11mcguire/mamba | | Version 1 |
| Software, algorithm | S-PrediXcan; S-MultiXcan | *Barbeira et al., 2018*; https://github.com/hakyimlab/MetaXcan | RRID:SCR_016739 | Version 1 |
| Software, algorithm | GTEx v8 mashr prediction models | https://predictdb.org/post/2021/07/21/gtex-v8-models-on-eqtl-and-sqtl/ | | |
| Other | GWAS Catalog | https://www.ebi.ac.uk/gwas/ | RRID:SCR_012745 | Section 'Definition of the genetic risk loci and putative functional impact' |

## GWAS in Latin Americans from SCOURGE

### The SCOURGE Latin American cohort

A total of 3729 COVID-19-positive cases were recruited across five countries from Latin America (Mexico, Brazil, Colombia, Paraguay, and Ecuador) by 13 participating centers (*Supplementary file 1*) from March 2020 to July 2021. In addition, we included 1082 COVID-19-positive individuals recruited between March and December 2020 in Spain who either had evidence of origin from a Latin American country or showed inferred genetic admixture between AMR, EUR, and AFR (with <0.05% SAS/EAS). These individuals were excluded from a previous SCOURGE study that focused on participants

with European genetic ancestries (*Cruz et al., 2022*). We used hospitalization as a proxy for disease severity and defined COVID-19-positive patients who underwent hospitalization as a consequence of the infection as cases and those who did not need hospitalization due to COVID-19 as controls.

Samples and data were collected with informed consent after the approval of the Ethics and Scientific Committees from the participating centers and by the Galician Ethics Committee Ref 2020/197. Recruitment of patients from IMSS (in Mexico City) was approved by the National Committee of Clinical Research from Instituto Mexicano del Seguro Social, Mexico (protocol R-2020-785-082).

Samples and data were processed following normalized procedures. The REDCap electronic data capture tool (*Harris et al., 2009*; *Harris et al., 2019*), hosted at Centro de Investigación Biomédica en Red (CIBER) from the Instituto de Salud Carlos III (ISCIII), was used to collect and manage demographic, epidemiological, and clinical variables. Subjects were diagnosed with COVID-19 based on quantitative PCR tests (79.3%) or according to clinical (2.2%) or laboratory procedures (antibody tests: 16.3%; other microbiological tests: 2.2%).

## SNP array genotyping

Genomic DNA was obtained from peripheral blood and isolated using the Chemagic DNA Blood 100 kit (PerkinElmer Chemagen Technologies GmbH), following the manufacturer's recommendations.

Samples were genotyped with the Axiom Spain Biobank Array (Thermo Fisher Scientific) following the manufacturer's instructions in the Santiago de Compostela Node of the National Genotyping Center (CeGen-ISCIII; http://www.usc.es/cegen). This array contains probes for genotyping a total of 757,836 SNPs. Clustering and genotype calling were performed using Axiom Analysis Suite v4.0.3.3 software.

## Quality control steps and variant imputation

A quality control (QC) procedure using PLINK 1.9 (*Purcell et al., 2007*) was applied to both samples and the genotyped SNPs. We excluded variants with a minor allele frequency (MAF) <1%, a call rate <98%, and markers strongly deviating from Hardy–Weinberg equilibrium expectations ($p<1 \times 10^{-6}$) with mid-p adjustment. We also explored the excess of heterozygosity to discard potential cross-sample contamination. Samples missing >2% of the variants were filtered out. Subsequently, we kept the autosomal SNPs, removed high-LD regions, and conducted LD pruning (windows of 1000 SNPs, with a step size of 80 and an $r^2$ threshold of 0.1) to assess kinship and estimate the global ancestral proportions. Kinship was evaluated based on IBD values, removing one individual from each pair with PI_HAT > 0.25 that showed a Z0, Z1, and Z2 coherent pattern (according to the theoretical expected values for each relatedness level). Genetic principal components (PCs) were calculated with PLINK with the subset of LD pruned variants.

Genotypes were imputed with the TOPMed version r2 reference panel (GRCh38) using the TOPMed Imputation Server, and variants with Rsq < 0.3 or with MAF <1% were filtered out. A total of 4348 individuals and 10,671,028 genetic variants were included in the analyses.

## Genetic admixture estimation

Global GIA, referred to the genetic similarity to the used reference individuals, was estimated with ADMIXTURE (*Alexander et al., 2009*) v1.3 software following a two-step procedure. First, we randomly sampled 79 European (EUR) and 79 African (AFR) samples from The 1000 Genomes Project (1KGP) (*Auton et al., 2015*) and merged them with the 79 Native American (AMR) samples from *Mao et al., 2007* keeping the biallelic SNPs. LD-pruned variants were selected from this merge using the same parameters as in the QC. We then ran an unsupervised analysis with K = 3 to redefine and homogenize the clusters and to compose a refined reference for the analyses by applying a threshold of ≥95% of belonging to a particular cluster. As a result, 20 AFR, 18 EUR, and 38 AMR individuals were removed. The same LD-pruned variants data from the remaining individuals were merged with the SCOURGE Latin American cohort to perform supervised clustering and estimate admixture proportions. A total of 471 samples from the SCOURGE cohort with >80% estimated European GIA were removed to reduce the weight of the European ancestral component, leaving a total of 3512 admixed Latin American (AMR) subjects for downstream analyses.

## Association analysis

The results for the SCOURGE Latin American GWAS were obtained by testing for COVID-19 hospitalization as a surrogate of severity. To accommodate the continuum of GIA in the cohort, we opted for a joint testing of all the individuals as a single study using a mixed regression model as this approach has demonstrated a greater power and sufficient control of population structure (*Wojcik et al., 2019*). The SCOURGE cohort consisted of 3512 COVID-19-positive patients: cases (n = 1625) were defined as hospitalized COVID-19 patients, and controls (n = 1887) were defined as non-hospitalized COVID-19-positive patients.

Logistic mixed regression models were fitted using the SAIGEgds (*Zheng and Davis, 2021*) package in R, which implements the two-step mixed SAIGE (*Zhou et al., 2018*) model methodology and the SPA test. Baseline covariables included sex, age (continuous), and the first 10 PCs. To account for potential heterogeneity in the recruitment and hospitalization criteria across the participating countries, we adjusted the models by groups of the recruitment areas classified into six categories: Brazil, Colombia, Ecuador, Mexico, Paraguay, and Spain. This dataset has not been used in any previously published GWAS of COVID-19.

## Meta-analysis of Latin American populations

The results of the SCOURGE Latin American cohort were meta-analyzed with the AMR HGI-B2 data, conforming our primary analysis. Summary results from the HGI freeze 7 B2 analysis corresponding to the admixed AMR population were obtained from the public repository (April 8, 2022: https://www.covid19hg.org/results/r7/), summing up 3077 cases and 66,686 controls from seven contributing studies. We selected the B2 phenotype definition because it offered more power, and the presence of population controls not ascertained for COVID-19 does not have a drastic impact on the association results.

The meta-analysis was performed using an inverse-variance weighting method in METAL (*Willer et al., 2010*). The average allele frequency was calculated, and variants with low imputation quality (Rsq < 0.3) were filtered out, leaving 10,121,172 variants for meta-analysis.

Heterogeneity between studies was evaluated with Cochran's *Q*-test. The inflation of results was assessed based on a genomic control (lambda).

## Replicability of associations

The model-based method MAMBA (*McGuire et al., 2021*) was used to calculate the posterior probabilities of replication for each of the lead variant (PPR; PP that an SNP has a non-zero replicable effect). We defined PPR <0.1 as a low posterior probability of replication, following the original paper, whereas those with a PPR >90% were considered consistent and likely to replicate in future studies. Variants with $p<1 \times 10^{-05}$ were clumped and combined with random pruned variants from the 1KGP AMR reference panel. Then, MAMBA was applied to the set of significant and non-significant variants.

Each of the lead variants was also tested for association with the main comorbidities in the SCOURGE cohort with logistic regression models (adjusted by the same base covariables as the GWAS).

## Definition of the genetic risk loci and putative functional impact

### Definition of lead variant and novel loci

To define the lead variants in the loci that were genome-wide significant, LD-clumping was performed on the meta-analysis data using a threshold $p$-value$<5 \times 10^{-8}$, clump distance = 1500 kb, independence set at a threshold $r^2 = 0.1$ and the SCOURGE cohort genotype data as the LD reference panel. Independent loci were deemed as a novel finding if they met the following criteria: (1) $p$-value$<5 \times 10^{-8}$ in the meta-analysis and $p$-value$>5 \times 10^{-8}$ in the HGI B2 ALL meta-analysis or in the HGI B2 AMR and AFR and EUR analyses when considered separately; (2) Cochran's *Q*-test for heterogeneity of effects is $<0.05/N_{loci}$, where $N_{loci}$ is the number of independent variants with $p<5 \times 10^{-8}$; and (3) the nearest gene has not been previously described in the latest HGIv7 update.

### Annotation and initial mapping

Functional annotation was performed with FUMA (*Watanabe et al., 2017*) for those variants with a $p$-value$<5 \times 10^{-8}$ or in moderate-to-strong LD ($r^2 > 0.6$) with the lead variants, where the LD was

calculated from the 1KGP AMR panel. Genetic risk loci were defined by collapsing LD blocks within 250 kb. Then, genes, scaled CADD v1.4 scores, and RegulomeDB v1.1 scores were annotated for the resulting variants with ANNOVAR in FUMA (*Watanabe et al., 2017*). Gene-based analysis was also performed using MAGMA (*de Leeuw et al., 2015*) as implemented in FUMA under the SNP-wide mean model using the 1KGP AMR reference panel. Significance was set at a threshold $p < 2.66 \times 10^{-6}$ (which assumes that variants can be mapped to a total of 18,817 genes).

FUMA allowed us to perform initial gene mapping by two approaches: (1) positional mapping, which assigns variants to genes by physical distance using 10 kb windows; and (2) eQTL mapping based on GTEx v.8 data from whole blood, lungs, lymphocytes, and esophageal mucosa tissues, establishing a false discovery rate (FDR) of 0.05 to declare significance for variant–gene pairs.

Subsequently, to assign the variants to the most likely gene driving the association, we refined the candidate genes by fine mapping the discovered regions.

## Bayesian fine-mapping

To conduct a Bayesian fine mapping, credible sets for the genetic loci considered novel findings were calculated on the results from each of the three meta-analyses to identify a subset of variants most likely containing the causal variant at the 95% confidence level, assuming that there is a single causal variant and that it has been tested. We used *corrcoverage* (https://cran.rstudio.com/web/packages/corrcoverage/index.html) for R to calculate the posterior probabilities of the variant being causal for all variants with an $r^2 > 0.1$ with the leading SNP and within 1 Mb except for the novel variant in chromosome 19, for which we used a window of 0.5 Mb. Variants were added to the credible set until the sum of the posterior probabilities was ≥0.95.

## VEP and V2G annotation

We used the Variant-to-Gene (V2G) score to prioritize the genes that were most likely affected by the functional evidence based on eQTL, chromatin interactions, in silico functional predictions, and distance between the prioritized variants and transcription start site (TSS), based on data from the Open Targets Genetics portal (*Ghoussaini et al., 2021*). Details of the data integration and the weighting of each of the datasets are described in detail at https://genetics-docs.opentargets.org/our-approach/data-pipeline. V2G is a score for ranking the functional genomics evidence that supports the connection between variants and genes (the higher the score the more likely the variant to be functionally implicated on the assigned gene). We used VEP release 111 (https://www.ensembl.org/info/docs/tools/vep/index.html; accessed April 10, 2024; *McLaren et al., 2016*) to annotate the following: gene symbol, function (exonic, intronic, intergenic, non-coding RNA, etc.), impact, feature type, feature, and biotype.

We queried the GWAS catalog (date of accession: 1/07/2024) for evidence of association of each of the prioritized genes with traits related to lung diseases or phenotypes. Lastly, those which were linked to COVID-19, infection, or lung diseases in the revised literature were classified as 'literature evidence'.

## Transcription-wide association studies

TWAS were conducted using the pretrained prediction models with MASHR-computed effect sizes on GTEx v8 datasets (*Barbeira et al., 2019a*; *Barbeira et al., 2021*). The results from the Latin American meta-analysis were harmonized and integrated with the prediction models through S-PrediXcan (*Barbeira et al., 2018*) for lung, whole blood, lymphocyte, and esophageal mucosal tissues. Statistical significance was set at p-value<0.05 divided by the number of genes that were tested for each tissue. Subsequently, we leveraged results for all 49 tissues and ran a multitissue TWAS (S-MultiXcan) to improve the power for association, as demonstrated recently (*Barbeira et al., 2019b*). TWAS was also performed using recently published gene expression datasets derived from a cohort of African Americans, Puerto Ricans, and Mexican Americans (GALA II-SAGE) (*Kachuri et al., 2023*).

## Cross-population meta-analyses

We conducted two additional meta-analyses to investigate the ability of combining populations to replicate our discovered risk loci. This methodology enabled the comparison of effects and the

significance of associations in the novel risk loci between the results from analyses that included or excluded other population groups.

The first meta-analysis comprised the five populations analyzed within HGI (B2-ALL). Additionally, to evaluate the three GIA components within the SCOURGE Latin American cohort (**Bryc et al., 2010**), we conducted a meta-analysis of the admixed AMR, EUR, and AFR cohorts (B2). All summary statistics were retrieved from the HGI repository. We applied the same meta-analysis methodology and filters as in the admixed AMR meta-analysis.

## Cross-population polygenic risk score

A PGS for critical COVID-19 was derived by combining the variants associated with hospitalization or disease severity that have been discovered to date. We curated a list of lead variants that were (1) associated with either severe disease or hospitalization in the latest HGIv7 release (**Kanai et al., 2023**) (using the hospitalization weights) or (2) associated with severe disease in the latest GenOMICC meta-analysis (**Pairo-Castineira et al., 2023**) that were not reported in the latest HGI release. A total of 48 markers were used in the PGS model (see **Supplementary file 13**) since two variants were absent from our study.

Scores were calculated and normalized for the SCOURGE Latin American cohort with PLINK 1.9. This cross-ancestry PGS was used as a predictor for hospitalization (COVID-19-positive patients who were hospitalized vs COVID-19-positive patients who did not necessitate hospital admission) by fitting a logistic regression model. Prediction accuracy for the PGS was assessed by performing 500 bootstrap resamples of the increase in the pseudo-$R^2$. We also divided the sample into deciles and percentiles to assess risk stratification. The models were fit for the dependent variable adjusting for sex, age, the first 10 PCs, and the sampling region (in the admixed AMR cohort) with and without the PGS, and the partial pseudo-$R^2$ was computed and averaged among the resamples.

A clinical severity scale was used in a multinomial regression model to further evaluate the power of this cross-ancestry PGS for risk stratification. These severity strata were defined as follows: (0) asymptomatic; (1) mild, that is, with symptoms, but without pulmonary infiltrates or need of oxygen therapy; (2) moderate, that is, with pulmonary infiltrates affecting <50% of the lungs or need of supplemental oxygen therapy; (3) severe disease, that is, with hospital admission and $PaO_2$ <65 mmHg or $SaO_2$ <90%, $PaO_2/FiO_2$<300, $SaO_2/FiO_2$<440, dyspnea, respiratory frequency ≥22 bpm, and infiltrates affecting >50% of the lungs; and (4) critical disease, that is, with an admission to the ICU or need of mechanical ventilation (invasive or noninvasive).

## Acknowledgements

The contribution of the Centro National de Genotipado (CEGEN) and Centro de Supercomputación de Galicia (CESGA) for funding this project by providing supercomputing infrastructures is also acknowledged. The authors are also particularly grateful for the supply of material and the collaboration of patients, health professionals from participating centers and biobanks. Namely, Biobanc-Mur, and biobancs of the Complexo Hospitalario Universitario de A Coruña, Complexo Hospitalario Universitario de Santiago, Hospital Clínico San Carlos, Hospital La Fe, Hospital Universitario Puerta de Hierro Majadahonda—Instituto de Investigación Sanitaria Puerta de Hierro—Segovia de Arana, Hospital Ramón y Cajal, IDIBGI, IdISBa, IIS Biocruces Bizkaia, IIS Galicia Sur. Also biobanks of the Sistema de Salud de Aragón, Sistema Sanitario Público de Andalucía, and Banco Nacional de ADN. Instituto de Salud Carlos III (COV20_00622 to AC, COV20/00792 to MB, COV20_00181 to CA, COV20_1144 to MAJS and AFR, PI20/00876 to CF); European Union (ERDF) 'A way of making Europe'. Fundación Amancio Ortega, Banco de Santander (to AC), Estrella de Levante SA and Colabora Mujer Association (to EGN) and Obra Social La Caixa (to RB); Agencia Estatal de Investigación (RTC-2017-6471-1 to CF), Cabildo Insular de Tenerife (CGIEU0000219140 'Apuestas científicas del ITER para colaborar en la lucha contra la COVID-19' to CF) and Fundación Canaria Instituto de Investigación Sanitaria de Canarias (PIFIISC20/57 to CF). SD-DA was supported by a Xunta de Galicia predoctoral fellowship.

# Additional information

## Funding

| Funder | Grant reference number | Author |
|---|---|---|
| Instituto de Salud Carlos III | COV20_00622 | Ángel Carracedo |
| Instituto de Salud Carlos III | COV20/00792 | Matilde Bustos |
| Instituto de Salud Carlos III | COV20_00181 | Carmen Ayuso |
| Instituto de Salud Carlos III | COV20_1144 | Amanda Fernández-Rodríguez<br>María A Jimenez-Sousa |
| Instituto de Salud Carlos III | PI20/00876 | Carlos Flores |
| European Regional Development Fund | A way of making Europe | Ángel Carracedo |
| Fundación Amancio Ortega | | Ángel Carracedo |
| Banco Santander | | Ángel Carracedo |
| Estrella de Levante S.A. | | Encarna Guillen-Navarro |
| Colabora Mujer Association | | Encarna Guillen-Navarro |
| Obra Social La Caixa | | Raúl C Baptista-Rosas |
| Agencia Estatal de Investigación | RTC-2017-6471-1 | Carlos Flores |
| Cabildo Insular de Tenerife | CGIEU0000219140 | Carlos Flores |
| Fundación Canaria Instituto de Investigación Sanitaria de Canarias | PIFIISC20/57 | Carlos Flores |
| Xunta de Galicia | | Silvia Diz-de Almeida |
| Axencia GAIN | | Silvia Diz-de Almeida |

The funders had no role in study design, data collection and interpretation, or the decision to submit the work for publication.

## Author contributions

Silvia Diz-de Almeida, Raquel Cruz, Conceptualization, Formal analysis, Validation, Investigation, Visualization, Methodology, Writing – original draft, Writing – review and editing; Andre D Luchessi, Formal analysis, Validation, Methodology, Writing – original draft, Writing – review and editing; José M Lorenzo-Salazar, Formal analysis, Methodology, Writing – original draft, Writing – review and editing; Miguel López de Heredia, Inés Quintela, Data curation, Project administration, Writing – review and editing; Rafaela González-Montelongo, Vivian Nogueira Silbiger, Marta Sevilla Porras, Jair Antonio Tenorio Castaño, Julian Nevado, Jose María Aguado, Carlos Aguilar, Sergio Aguilera-Albesa, Virginia Almadana, Berta Almoguera, Nuria Alvarez, Álvaro Andreu-Bernabeu, Eunate Arana-Arri, Celso Arango, María J Arranz, Maria-Jesus Artiga, Raúl C Baptista-Rosas, María Barreda- Sánchez, Moncef Belhassen-Garcia, Joao F Bezerra, Marcos AC Bezerra, Lucía Boix-Palop, María Brion, Ramón Brugada, Matilde Bustos, Enrique J Calderón, Cristina Carbonell, Luis Castano, Jose E Castelao, Rosa Conde-Vicente, M Lourdes Cordero-Lorenzana, Jose L Cortes-Sanchez, Marta Corton, M Teresa Darnaude, Victor del Campo-Pérez, Aranzazu Diaz de Bustamante, Elena Domínguez-Garrido, Rocío Eirós, María Carmen Fariñas, María J Fernandez-Nestosa, Uxía Fernández-Robelo, Amanda Fernández-Rodríguez, Tania Fernández-Villa, Manuela Gago-Dominguez, Belén Gil-Fournier, Javier Gómez-Arrue, Beatriz González Álvarez, Fernan Gonzalez Bernaldo de Quirós, Anna González-Neira, Javier González-Peñas, Juan F Gutiérrez-Bautista, María José Herrero, Antonio Herrero-Gonzalez, María A Jimenez-Sousa, María Claudia Lattig, Anabel Liger Borja, Rosario Lopez-Rodriguez, Esther Mancebo, Caridad Martín-López, Vicente Martín, Oscar Martinez-Nieto, Iciar Martinez-Lopez, Michel

F Martinez-Resendez, Angel Martinez-Perez, Juliana F Mazzeu, Eleuterio Merayo Macías, Pablo Minguez, Victor Moreno Cuerda, Silviene F Oliveira, Eva Ortega-Paino, Mara Parellada, Estela Paz-Artal, Ney PC Santos, Patricia Perez, M Elena Pérez-Tomás, Teresa Perucho, Mellina Pinsach-Abuin, Guillermo Pita, Ericka N Pompa-Mera, Gloria L Porras-Hurtado, Aurora Pujol, Soraya Ramiro León, Salvador Resino, Marianne R Fernandes, Emilio Rodríguez-Ruiz, Fernando Rodriguez-Artalejo, José A Rodriguez-Garcia, Francisco Ruiz-Cabello, Javier Ruiz-Hornillos, Pablo Ryan, José Manuel Soria, Juan Carlos Souto, Eduardo Tamayo, Alvaro Tamayo-Velasco, Juan Carlos Taracido-Fernandez, Alejandro Teper, Lilian Torres-Tobar, Miguel Urioste, Juan Valencia-Ramos, Zuleima Yáñez, Ruth Zarate, Itziar de Rojas, Agustín Ruiz, Pascual Sánchez, Luis Miguel Real, Encarna Guillen-Navarro, Carmen Ayuso, Data curation, Writing – review and editing; Alba De Martino-Rodríguez, Patricia Pérez-Matute, Data curation; SCOURGE Cohort Group, Data curation, Methodology, Resources; Esteban Parra, Investigation, Methodology, Writing – review and editing; José A Riancho, Conceptualization, Supervision, Funding acquisition, Investigation, Methodology, Project administration, Writing – review and editing; Augusto Rojas-Martinez, Pablo Lapunzina, Conceptualization, Supervision, Funding acquisition, Investigation, Project administration, Writing – review and editing; Carlos Flores, Ángel Carracedo, Conceptualization, Supervision, Funding acquisition, Investigation, Methodology, Writing – original draft, Project administration, Writing – review and editing

### Author ORCIDs
Silvia Diz-de Almeida http://orcid.org/0000-0003-2813-8928
Vivian Nogueira Silbiger https://orcid.org/0000-0002-9252-0278
Raúl C Baptista-Rosas https://orcid.org/0000-0002-0273-4740
Alba De Martino-Rodríguez https://orcid.org/0000-0002-2436-9852
María J Fernandez-Nestosa https://orcid.org/0000-0001-7764-7791
María Claudia Lattig https://orcid.org/0000-0003-2113-9266
Angel Martinez-Perez https://orcid.org/0000-0002-5934-1454
Silviene F Oliveira https://orcid.org/0000-0002-7741-0257
M Elena Pérez-Tomás https://orcid.org/0000-0002-5429-3414
Aurora Pujol https://orcid.org/0000-0002-9606-0600
Pablo Ryan https://orcid.org/0000-0002-4212-7419
Luis Miguel Real https://orcid.org/0000-0003-4932-7429
Esteban Parra https://orcid.org/0000-0002-2057-8577
José A Riancho https://orcid.org/0000-0003-0691-8755
Augusto Rojas-Martinez https://orcid.org/0000-0003-3765-6778
Carlos Flores https://orcid.org/0000-0001-5352-069X
Ángel Carracedo https://orcid.org/0000-0003-1085-8986

### Ethics
Human subjects: Ethics and Scientific Committees from the participating centres and by the Galician Ethics Committee Ref 2020/197 gave ethical approval for this work. Recruitment of patients from IMSS (in Mexico City), was approved by of the National Committee of Clinical Research, Instituto Mexicano del Seguro Social, Mexico (protocol R-2020-785-082).

Reviewer #1 (Public review): https://doi.org/10.7554/eLife.93666.3.sa1
Reviewer #2 (Public review): https://doi.org/10.7554/eLife.93666.3.sa2
Reviewer #3 (Public review): https://doi.org/10.7554/eLife.93666.3.sa3
Author response https://doi.org/10.7554/eLife.93666.3.sa4

## Additional files

### Supplementary files
• Supplementary file 1. Participating centers.

• Supplementary file 2. Independent variants with p-value<$1 \times 10^{-05}$ in the SC-HGI_AMR GWAS meta-analysis (hg38). EA: effect allele; NEA: non-effect allele; EAF: effect allele frequency; EAF_avg: averaged effect allele frequency; FreqSE: standard error of averaged effect allele frequency; SCOURGE_AMR: SCOURGE Latin-America; HGIB2_AMR: HGI meta-analysis of AMR studies.

• Supplementary file 3. Annotated SNPs in moderate-to-strong LD with lead SNPs of the genome-

wide significant loci in the SC-HGI_AMR GWAS meta-analysis, with ANNOVAR. NEA: non-effect allele; EA: effect allele; $r^2$: maximum $r^2$ of the SNP with one of the independent SNPs; IndSigSNP: the independent SNP which has the maximum $r^2$ value with the SNP; dist: distance to the nearest gene; func: functional consequence of the SNP on the gene; CADD: CADD score; RDB: RegulomeDB score; minChrState: the minimum 15-core chromatin state across 127 tissues/cell types; commonChrState: the most common 15-core chromatin state across 127 tissues/cell types; posMapFilt: 1 if the SNP was used for positional mapping, 0 otherwise; eqtlMapFilt: 1 if the SNP was used for eQTL mapping, 0 otherwise.

• Supplementary file 4. Results from the MAGMA gene-based analysis in the SC-HGI_AMR GWAS meta-analysis (hg37). NSNPS: number of SNPs in the gene; NPARAM: the number of relevant parameters used in the model; ZSTAT: z statistics.

• Supplementary file 5. Prioritized genes by eQTL and positional mapping by FUMA in the SC-HGI_AMR GWAS meta-analysis results (hg37). HUGO: HGNC gene symbol; pLI: pLI score from ExAC database, probability of being intolerant to loss of function (higher the score, higher the intolerance); ncRVIS: non-coding residual variation intolerance score (higher the score, higher intolerance to non-coding variation); posMapSNPs: number of SNPs mapped by positional mapping; posMapMaxCADD: the maximum CADD score of mapped SNPs by positional mapping; eqtlMapSNPS: the number of SNPs mapped to the genes based on eQTL mapping; eqtlMapminP: the minimum eQTL p-value of mapped SNPs; eqtlMapminQ: the minimum eQTL FDR of mapped SNPs; eqtlMapts: tissue of mapped eQTLs; eqtlDirection: consequential direction of mapped eQTL SNPs after aligning the risk alleles; minGwasP: minimum GWAS p-value of mapped eQTLs; IndSigSNPs: independent SNPs that are in LD with the mapped SNPs.

• Supplementary file 6. Fine-mapped credible set derived with corrcoverage (95%) for the associated region in chromosome 2 (BAZ2B).

• Supplementary file 7. VEP annotations for the variants included in the fine-mapped credible sets for the novel associated loci in chromosome 2 (hg38).

• Supplementary file 8. V2G scores for the variants included in the fine-mapped credible sets in the novel risk loci from chromosomes 2 and 16 (hg38). Shaded in green, the prioritized gene by the V2G score.

• Supplementary file 9. MultiXcan results for the SC-HGI_AMR GWAS meta-analysis. N: number of tissues available for the gene; n_indep: number of independent components of variation kept among the tissues' predictions; p_i_best: best p-value of single tissue S-prediXcan association; t_i_best: name of best single tissue S-prediXcan association; p_i_worst: worst p-value of single tissue S-prediXcan association; t_i_worst: name of worst single tissue S-prediXcan association; eigen_max: eigenvalue of the top independent component in the SVD decomposition of predicted expression correlation; eigen_min: eigenvalue of the last independent component in the SVD decomposition of predicted expression correlation; eigen_min_kept: eigenvalue of the smallest independent component that was kept in the SVD decomposition of predicted expression correlation; z_min: minimum z-score among single-tissue S-prediXcan associations; z_max: maximum z-score among single-tissue S-prediXcan associations; z_mean: mean z-score among single tissue S-prediXcan associations; z_sd: standard deviation of the mean z-score among single-tissue S-prediXcan associations; tmi: trace of T*T', where T is the correlation of predicted expression levels for different tissues multiplied by its SVD pseudo-inverse and is an estimate for the number of independent components of variation in predicted expression across tissues.

• Supplementary file 10. Top 10 genes for the TWAS trained with the GALA II-SAGE models in admixed Americans. Bonferroni correction thresholds: Pooled p<4.19E-06; PR p<4.99E-06; MX p<5.19E-06; AA p<4.67E-06. Var_g: variance of the gene expression; pred_perf_r2: cross-validated $R^2$ of tissue model's correlation to gene's measured transcriptome; pref_perf_qval: qval of tissue model's correlation to gene's measured transcriptome; n_snps_used: number of snps from GWAS used in S-prediXcan analysis; n_snp_in_cov: number of snps in the covariance matrix; n_snps_in_model: number of snps in the model; best_gwas_p: the highest p-value from GWAS snps used in this model; largest_weight: the largest weight in this model.

• Supplementary file 11. Independent variants with p-value<1e-05 in the SC-HGI_ALL GWAS meta-analysis (hg38). EA: effect allele; NEA: non-effect allele; EAF_avg: averaged effect allele frequency; FreqSE: standard error of averaged effect allele frequency.

• Supplementary file 12. Results of the 40 lead variants associated with COVID-19 hospitalization in the HGIv7 (hg38). SC-HGI_ALL: meta-analysis SCOURGE-HGI_ALL; SC-HGI_AMR: meta-analysis SCOURGE-HGI_AMR; SC-HGI_3POP: meta-analysis SCOURGE-HGI_3POP.

• Supplementary file 13. Independent variants with p-value<1e-05 in the SC-HGI_3POP GWAS meta-analysis (hg38). EA: effect allele; NEA: non-effect allele; EAF_avg: average effect allele frequency; FreqSE: standard error of averaged effect allele frequency.

• Supplementary file 14. Instruments used in the polygenic risk score model (hg38).

• Supplementary file 15. Multinomial regression results. Reference class for the multinomial regression is 'asymptomatic'.

• MDAR checklist

## Data availability

Summary statistics from the SCOURGE Latin American GWAS and the analysis scripts are available from the public repository https://github.com/CIBERER/Scourge-COVID19 (copy archived at *CIBERER, 2024*).

The following previously published datasets were used:

| Author(s) | Year | Dataset title | Dataset URL | Database and Identifier |
|---|---|---|---|---|
| Consortium GTEx | 2020 | GTEx V8 | https://www.gtexportal.org/home/downloads/adult-gtex/qtl | GTEx Portal, Single-Tissue cis-QTL |
| Consortium Genomes Project | 2016 | 1000 Genomes Phase 3 | https://www.cog-genomics.org/plink/2.0/resources | PLINK 2.0 Resources, 1000 Genomes phase 3 |
| COVID-19 Host Genetics Initiative | 2022 | COVID19-hg GWAS meta-analyses round 7 | https://www.covid19hg.org/results/r7/ | COVID-19 hg repository, r7 |

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
