## [Editor Report · eLife assessment]

The authors conducted a **valuable** GWAS meta-analysis for COVID-19 hospitalization in admixed American populations and prioritized risk variants and genes. The evidence supporting the claims of the authors is **solid**. The work will be of interest to scientists studying the genetic basis of COVID pathogenesis.

---

## [Referee Report · Reviewer #1 (Public review)]

Summary:

This paper conducted a GWAS meta-analysis for COVID-19 hospitalization among admixed American populations. The authors identified four genome-wide significant associations, including two novel loci (BAZ2B and DDIAS), and an additional risk locus near CREBBP using cross-ancestry meta-analysis. They utilized multiple strategies to prioritize risk variants and target genes. Finally, they constructed and assessed a polygenic risk score model with 49 variants associated with critical COVID-19 conditions.

Strengths:

Given that most of the previous studies were done in European ancestries, this study provides unique findings about the genetics of COVID-19 in admixed American populations. The GWAS data would be a valuable resource for the community. The authors conducted comprehensive analyses using multiple different strategies, including Bayesian fine mapping, colocalization, TWAS, etc., to prioritize risk variants and target genes. The polygenic risk score (PGS) result demonstrated the ability of cross-population PGS model for COVID-19 risk stratification.

Weaknesses:

(1) One of the major limitations of this study is that the GWAS sample size is relatively small, which limits its power.

(2) Lack of replication cohort.

(3) Colocalization and TWAS used eQTL data from GTEx data, which are mainly from European ancestries.

Comments on latest version:

The authors addressed most of my concerns.

---

## [Referee Report · Reviewer #2 (Public review)]

This is a genome-wide association study of COVID-19 in individuals of admixed American ancestry (AMR) recruited from Brazil, Colombia, Ecuador, Mexico, Paraguay and Spain. After quality control and admixture analysis, a total of 3,512 individuals were interrogated for 10,671,028 genetic variants (genotyped + imputed). The genetic association results for these cohorts were meta-analyzed with the results from The Host Genetics Initiative (HGI), involving 3,077 cases and 66,686 controls. The authors found two novel genetic loci associated with COVID-19 at 2q24.2 (rs13003835) and 11q14.1 (rs77599934), and other two independent signals at 3p21.31 (rs35731912) and 6p21.1 (rs2477820) already reported as associated with COVID-19 in previous GWASs. Additional meta-analysis with other HGI studies also suggested risk variants near CREBBP, ZBTB7A and CASC20 genes.

Strengths:

These findings rely on state-of-the-art methods in the field of Statistical Genomics and help to address the issue of low number of GWASs in non-European populations, ultimately contributing to reduce health inequalities across the globe.

Weaknesses:

There is no replication cohort, as acknowledged by the authors (page 29, line 587) and no experimental validation to assess the biological effect of putative causal variants/genes. Thus, the study provides good evidence of association, rather than causation, between the genetic variants and COVID-19.

Comments on latest version:

The issues identified in the first round of review were well addressed by the authors in the revised version of the manuscript.

---

## [Referee Report · Reviewer #3 (Public review)]

Summary:

In the context of the SCOURGE consortium's research, the authors conduct a GWAS meta-analysis on 4,702 hospitalized individuals of admixed American descent suffering from COVID-19. This study identified four significant genetic associations, including two loci initially discovered in Latin American cohorts. Furthermore, a trans-ethnic meta-analysis highlighted an additional novel risk locus in the CREBBP gene, underscoring the critical role of genetic diversity in understanding the pathogenesis of COVID-19.

Strengths:

(1) The study identified two novel severe COVID-19 loci (BAZ2B and DDIAS) by the largest GWAS meta-analysis for COVID-19 hospitalization in admixed Americans.

(2) With a trans-ethnic meta-analysis, an additional risk locus near CREBBP was identified.

Weaknesses:

(1) The GWAS power is limited due to the relatively small number of cases.

(2) There is no replication study for the novel severe COVID-19 loci, which may lead to false positive findings.

(3) The variants selected for the PGS appear arbitrary and may not leverage the GWAS findings.

(4) The TWAS models were predominantly trained on European samples, and there is no replication study for the findings as well.

---

## [Author Response]

The following is the authors’ response to the original reviews.

**Public Reviews:**

**Reviewer #1 (Public Review):**
Summary:This paper conducted a GWAS meta-analysis for COVID-19 hospitalization among admixed American populations. The authors identified four genome-wide significant associations, including two novel loci (BAZ2B and DDIAS), and an additional risk locus near CREBBP using cross-ancestry meta-analysis. They utilized multiple strategies to prioritize risk variants and target genes. Finally, they constructed and assessed a polygenic risk score model with 49 variants associated with critical COVID-19 conditions.Strengths:Given that most of the previous studies were done in European ancestries, this study provides unique findings about the genetics of COVID-19 in admixed American populations. The GWAS data would be a valuable resource for the community. The authors conducted comprehensive analyses using multiple different strategies, including Bayesian fine mapping, colocalization, TWAS, etc., to prioritize risk variants and target genes. The polygenic risk score (PGS) result demonstrated the ability of the cross-populationPGS model for COVID-19 risk stratification.

Thank you very much for the positive comments and the willingness to revise this manuscript.

Weaknesses:(1) One of the major limitations of this study is that the GWAS sample size is relatively small, which limits its power.(2) The fine mapping section is unclear and there is a lack of information. The authors assumed one causal signal per locus, and only provided credible sets, but did not provide posterior inclusion probabilities (PIP) for the variants to be causal.(3) Colocalization and TWAS used eQTL data from GTEx data, which are mainly from European ancestries. It is unclear how much impact the ancestry mismatch would have on the result. The readers should be cautious when interpreting the results and designing follow-up studies.

We agree with that the sample size is relatively small. Despite that, it was sufficient to reveal novel risk loci supporting the robustness of the main findings. We have indicated this limitation at the end of the discussion section.

Thank you for rising this point. As suggested, we have also used SuSIE, which allows to assume more than one causal signal per locus. However, in this case the results were not different from those obtained with the original Bayesian colocalization performed with corrcoverage. Regarding the PIP, at the fine mapping stage we are inclined to put more weight on the functional annotations of the variants in the credible set than on the statistical contributions to the signal. This is the reason why we prefer not to put weight on the PIP of the variants but prioritize variants that were enriched functional annotations.

This is a good point regarding the lack of diversity in GTEx data. We have also used data from AMR populations (GALA II-SAGE models), although it was only available for blood tissue. Regarding the ancestry mismatch between datasets, several studies have attempted to explore the impact. Gay et al. (PMID: 32912333) studied local ancestry effects on eQTLs from the GTEx consortium and concluded that adjustment of eQTLs by local ancestry only yields modest improvement over using global ancestry (as done in GTEx). Moreover, the colocalization results between adjusting by Local Ancestry and Global Ancestry were not significantly different. Besides, Mogil et al. (PMID: 30096133) observed that genes with higher heritability share genetic architecture between populations. Nevertheless, both studies have evidenced decreased power and poorer predictive performances regarding gene expression because of reduced diversity in eQTL analyses. As consequence of the ancestry mismatch, we now warn the readers that this may compromise signal detection (Discussion, lines 531-533).

**Reviewer #2 (Public Review):**
This is a genome-wide association study of COVID-19 in individuals of admixed American ancestry (AMR) recruited from Brazil, Colombia, Ecuador, Mexico, Paraguay, and Spain. After quality control and admixture analysis, a total of 3,512 individuals were interrogated for 10,671,028 genetic variants (genotyped + imputed). The genetic association results for these cohorts were meta-analyzed with the results from The Host Genetics Initiative (HGI), involving 3,077 cases and 66,686 controls. The authors found two novel genetic loci associated with COVID-19 at 2q24.2 (rs13003835) and 11q14.1 (rs77599934), and other two independent signals at 3p21.31 (rs35731912) and 6p21.1 (rs2477820) already reported as associated with COVID-19 in previous GWASs. Additional meta-analysis with other HGI studies also suggested risk variants near CREBBP, ZBTB7A, and CASC20 genes.Strengths:These findings rely on state-of-the-art methods in the field of Statistical Genomics and help to address the issue of a low number of GWASs in non-European populations, ultimately contributing to reducing health inequalities across the globe.

Thank you very much for the positive comments and the willingness to revise this manuscript.

Weaknesses:There is no replication cohort, as acknowledged by the authors (page 29, line 587), and no experimental validation to assess the biological effect of putative causal variants/genes. Thus, the study provides good evidence of association, rather than causation, between the genetic variants and COVID-19. Lastly, I consider it crucial to report the results for the SCOURGE Latin American GWAS, in addition to its meta-analysis with HGI results, since HGI data has a different phenotype scheme (Hospitalized COVID vs Population) compared to SCOURGE (Hospitalized COVID vs Non-hospitalized COVID).

We essentially agree with the reviewer in that one of the main limitations of the study is the lack of a replication stage because of the use of all available datasets on a one-stage analysis. To contribute to the interpretation of the findings in the absence of a replication stage, we now assessed the replicability of the novel loci using the Meta-Analysis Model-based Assessment of replicability (MAMBA) approach (PMID: 33785739) and included the posterior probabilities of replication in Table 2. We also explored further the potential replicability of signals in other populations. We agree that the results should be interpreted in terms of associations given the lack of functional validation of main findings, so we have slightly modified the discussion.

As suggested, the SCOURGE Latin American GWAS summary is now accessible by direct request to the Consortium GitHub repository (https://github.com/CIBERER/Scourge-COVID19) (lines 797-799). We have also included the results from the SCOURGE GWAS analysis for the replication of the 40 lead variants in the Supplementary Table 12. Results from the SCOURGE GWAS for the lead variants in the AMR meta-analysis with HGI were already included in the Supplementary Table 2. As note, we have not been able to conduct the meta-analysis with the same hospitalization scheme as in the HGI study since the population-specific results for those analyses were not publicly released. However, sensitivity analyses included within the supplementary material from the COVID-19 Host Genetics Initiative (2021) stated that there were no significant differences in effects (Odds Ratios) between analyses using population controls or just non-hospitalized COVID-19 patients.

**Reviewer #3 (Public Review):**
Summary:In the context of the SCOURGE consortium's research, the authors conduct a GWAS meta-analysis on 4,702 hospitalized individuals of admixed American descent suffering from COVID-19. This study identified four significant genetic associations, including two loci initially discovered in Latin American cohorts. Furthermore, a trans-ethnic meta-analysis highlighted an additional novel risk locus in the CREBBP gene, underscoring the critical role of genetic diversity in understanding the pathogenesis of COVID-19.Strengths:(1) The study identified two novel severe COVID-19 loci (BAZ2B and DDIAS) by the largest GWAS meta-analysis for COVID-19 hospitalization in admixed Americans.(2) With a trans-ethnic meta-analysis, an additional risk locus near CREBBP was identified.

Thank you very much for the positive comments and the willingness to revise this manuscript.

Weaknesses:(1) The GWAS power is limited due to the relatively small number of cases.(2) There is no replication study for the novel severe COVID-19 loci, which may lead to false positive findings.

We agree with that the sample size is relatively small. Despite that, it was sufficient to reveal novel risk loci supporting the robustness of the main findings. We have indicated this limitation at the end of the discussion section.

Regarding the lack of a replication study, we now assessed the replicability of the novel loci using the Meta-Analysis Model-based Assessment of replicability (MAMBA) approach (PMID: 33785739). We have included the posterior probabilities of replication in Table 2.

(3) Significant differences exist in the ages between cases and controls, which could potentially introduce biased confounders. I'm curious about how the authors treated age as a covariate. For instance, did they use ten-year intervals? This needs clarification for reproducibility.

Thank you for rising this point. Age was included as a continuous variable. This has been now indicated in line 667 (within Material and Methods).

(4)"Those in the top PGS decile exhibited a 5.90-fold (95% CI=3.29-10.60, p=2.79x10-9) greater risk compared to individuals in the lowest decile". I would recommend comparing with the 40-60% PGS decile rather than the lowest decile, as the lowest PGS decile does not represent 'normal controls'.

Thank you. In the revised version, the PGS categories was compared following the recommendation (lines 461-463).

(5) In the field of PGS, it's common to require an independent dataset for training and testing the PGS model. Here, there seems to be an overfitting issue due to using the same subjects for both training and testing the variants.

We are sorry for the misunderstanding. In fact, we have followed the standard to avoid overfitting of the PGS model and have used different training and testing datasets. The training data (GWAS) was the HGI-B2 ALL meta-analysis, in which our AMR GWAS was not included. The PRS model was then tested in the SCOURGE AMR cohort. However, it is true that we did test the combination of the PRS adding the new discovered variants in the SCOURGE cohort. To avoid potential overfitting by adding the new loci, we have excluded from the manuscript the results on which we included the newly discovered variants.

(6) The variants selected for the PGS appear arbitrary and may not leverage the GWAS findings without an independent training dataset.

Again, we are sorry for the misunderstanding. The PGS model was built with 43 variants associated with hospitalization or severity within the HGI v7 results and 7 which were discovered by the GenOMICC consortium in their latest study and were not in the latest HGI release. The variants are included within the Supplementary Table 14, but we have now annotated the discovery GWAS.

(7) The TWAS models were predominantly trained on European samples, and there is no replication study for the findings as well.

This is a good point regarding the lack of diversity in GTEx data. We have also used data from AMR populations (GALA II-SAGE models), although it was only available for blood tissue. Regarding the ancestry mismatch between datasets, several studies have attempted to explore the impact. Gay et al. (PMID: 32912333) studied local ancestry effects on eQTLs from the GTEx consortium and concluded that adjustment of eQTLs by local ancestry only yields modest improvement over using global ancestry (as done in GTEx). Moreover, the colocalization results between adjusting by Local Ancestry and Global Ancestry were not significantly different. Besides, Mogil et al. (PMID: 30096133) observed that genes with higher heritability share genetic architecture between populations. Nevertheless, both studies have evidenced decreased power and poorer predictive performances regarding gene expression because of reduced diversity in eQTL analyses. As consequence of the ancestry mismatch, we now warn the readers that this may compromise signal detection (Discussion, lines 531-533).

**Recommendations for the authors:**

**Reviewer #1 (Recommendations For The Authors):**
(1) The authors mentioned the fine mapping method did not converge for the locus in chr 11. I would consider trying a different fine-mapping method (such as SuSiE or FINEMAP). It would be helpful to provide posterior inclusion probabilities (PIP) for the variants in fine mapping results and plot the PIP values in the regional association plots.

As suggested, we have also used SuSIE, which allows to assume more than one causal signal per locus. However, in this case the results were not different from those obtained with the original Bayesian colocalization performed with corrcoverage. SuSIE’s fine-mapping for chromosome 11 prioritized a single variant, which is likely due to the rare frequency. Thus, we have maintained the fine-mapping as it was originally indicated in the previous version of the manuscript but have now included the credible set in Supplementary Table 6.

Regarding the PIP, at the fine mapping stage we are inclined to put more weight on the functional annotations of the variants in the credible set than on the statistical contributions to the signal. This is the reason why we prefer not to put weight on the PIP of the variants but prioritize variants that were enriched functional annotations.

(2) Please provide more detailed information about the VEP and V2G analysis and how to interpret those results. My understanding of V2G is that it includes different sources of information (such as molecular QTLs and chromatin interactions from different tissues/cell types, etc.). It is unclear what sources of information and weight settings were used in the V2G model.

Thank you for rising this point. As suggested, we have clarified the basis for VEP and V2G and the interpretation (lines 732-743).

(3) The authors identified multiple genes with different strategies, e.g. FUMA, V2G, COLOC, TWAS, etc. How many genes were found/supported by evidence provided by multiple methods? It could be helpful to have a table summarizing the risk genes found by different strategies, and the evidence supporting the genes. e.g. which genes are found by which methods, and the biological functions of the genes, etc.

Thank you for rising this point. As suggested, we now added a new figure (Figure 5) to summarize the findings with the multiple methods used.

(4) It would be helpful to make the code/scripts available for reproducibility.

As suggested, the SCOURGE Latin American GWAS summary and the analysis scripts (https://github.com/CIBERER/Scourge-COVID19/tree/main/scripts/novel-risk-hosp-AMR-2024) are now accessible in the Consortium GitHub repository (https://github.com/CIBERER/Scourge-COVID19) (lines 806-807).

(5) The fonts in some of the figures (e.g. Figure 2) are hard to read.

Thank you. We have now included the figures as SVG files.

**Reviewer #2 (Recommendations For The Authors):**
- The abstract lacks a conclusion sentence.

Thank you. As suggested, we have included two additional sentences with broad conclusions from the study. We preferred to avoid relying on conclusions related to known or new biological links of the prioritized genes given the lack of functional validation of main findings.

- Regarding the association analysis (page 27, line 677), I wonder if some of the 10 principal components (PCs) are capturing information about the recruitment areas (countries). It may be relevant to test for multicollinearity among these variables.

Since we acknowledge that some of the categories might be correlated with a certain PC but not all of them do, we have calculated GVIF values for the main variables to assess the categorical variable as a single entity. The scaled GVIF^1(1/2*Df) value for the categorical variable is 1.52. Thus, if we square this value, we obtain 2.31, which can be then used for applying usual rule-of-thumb for VIF values.

- Still on the topic of association analysis, did the authors adjust the logistic model for comorbidities variables from Table 1? Given these comorbidities also have a genetic component and their distribution differs between non-hospitalized vs hospitalized, I am concerned that comorbidities might be confounding the association between genetic variants and COVID.

We did not adjust by comorbidities since HGI studies were not adjusted either and we aimed to be as aligned as possible with HGI. However, as suggested, we have now tested the association between each of the comorbidities in Table 1 and each of the variants in Table 2, using the comorbidities as dependent variables and adjusting for the main covariables (age, sex, PCs and country of recruitment). None of the variants were significantly associated to the comorbidities (line 333).

- If I understood correctly, the 49 genetic variants used to develop the polygenic risk score model (PRS) were based on the HGI total sample size (data release 7), which is predominantly of European ancestry. I am concerned about the prediction accuracy in the AMR population (PRS transferability issue).

We have explored literature in search of other PRS to compare the associated OR in our cohort with ORs calculated in European populations. Horowitz et al. (2022) reported an OR of 1.38 for the top 10% with respect to hospitalization risk in European individuals using a GRS with 12 variants.

We acknowledge that this might be an issue and is now explained in discussion of the revised version (lines 561-568). However, as this is the first time a PRS for COVID-19 is applied to a relatively large AMR cohort, we believe that this analysis will be of value for further analyses regarding PRS transferability, providing a source for comparison in further studies.

- On page 23, line 579, the authors acknowledge their "GWAS is underpowered". This sentence requires a sample/power calculation, otherwise, I suggest using "is likely underpowered".

Thanks for the input. We have modified the sentence as suggested.

**Reviewer #3 (Recommendations For The Authors):**
I wonder if the authors have an approximate date when the GWAS summary statistic will be available. I reviewed some manuscripts in the past, and the authors claimed they would deposit the data soon, but in fact it would not happen until 2 years later.

The summary statistics are already available from the SCOURGE Consortium repository https://github.com/CIBERER/Scourge-COVID19 (lines 806-807).